# Genetic and Diet-Induced Animal Models for Non-Alcoholic Fatty Liver Disease (NAFLD) Research

**DOI:** 10.3390/ijms232415791

**Published:** 2022-12-13

**Authors:** Christina-Maria Flessa, Narjes Nasiri-Ansari, Ioannis Kyrou, Bianca M. Leca, Maria Lianou, Antonios Chatzigeorgiou, Gregory Kaltsas, Eva Kassi, Harpal S. Randeva

**Affiliations:** 1Department of Biological Chemistry, Medical School, National and Kapodistrian University of Athens, 11527 Athens, Greece; 2Warwickshire Institute for the Study of Diabetes, Endocrinology and Metabolism (WISDEM), University Hospitals Coventry and Warwickshire NHS Trust, Coventry CV2 2DX, UK; 3Warwick Medical School, University of Warwick, Coventry CV4 7AL, UK; 4Research Institute for Health and Wellbeing, Coventry University, Coventry CV1 5FB, UK; 5Aston Medical School, College of Health and Life Sciences, Aston University, Birmingham B4 7ET, UK; 6Laboratory of Dietetics and Quality of Life, Department of Food Science and Human Nutrition, School of Food and Nutritional Sciences, Agricultural University of Athens, 11855 Athens, Greece; 7Department of Physiology, Medical School, National and Kapodistrian University of Athens, 11527 Athens, Greece; 8Endocrine Unit, 1st Department of Propaedeutic Internal Medicine, Laiko Hospital, National and Kapodistrian University of Athens, 11527 Athens, Greece

**Keywords:** non-alcoholic fatty liver disease, NAFLD, non-alcoholic steatohepatitis, NASH, animal models, mouse, cirrhosis

## Abstract

A rapidly increasing incidence of non-alcoholic fatty liver disease (NAFLD) is noted worldwide due to the adoption of western-type lifestyles and eating habits. This makes the understanding of the molecular mechanisms that drive the pathogenesis of this chronic disease and the development of newly approved treatments of utmost necessity. Animal models are indispensable tools for achieving these ends. Although the ideal mouse model for human NAFLD does not exist yet, several models have arisen with the combination of dietary interventions, genetic manipulations and/or administration of chemical substances. Herein, we present the most common mouse models used in the research of NAFLD, either for the whole disease spectrum or for a particular disease stage (e.g., non-alcoholic steatohepatitis). We also discuss the advantages and disadvantages of each model, along with the challenges facing the researchers who aim to develop and use animal models for translational research in NAFLD. Based on these characteristics and the specific study aims/needs, researchers should select the most appropriate model with caution when translating results from animal to human.

## 1. Introduction

Non-alcoholic fatty liver disease (NAFLD) is the hepatic manifestation of a cluster of conditions associated with metabolic dysfunction and is characterized by the accumulation of fat in the liver [1]. The global prevalence of NAFLD is constantly increasing and is estimated to be around 25% in the general population [2]. Histopathologically, NAFLD begins with the development of steatosis (accumulation of fat in the form of triglycerides in hepatocytes) with or without mild inflammation (non-alcoholic fatty liver, NAFL) and may progress to non-alcoholic steatohepatitis (NASH) characterized by variable degrees of mainly macrovesicular steatosis, necroinflammation with enlarged and rounded (ballooned) hepatocytes, the development of cytoplasmic inclusions (Mallory-Denk bodies), variable and mixed inflammatory infiltrate and the development of fibrosis with a predominantly perivenular and pericellular distribution [3,4]. Prolonged inflammation with increased oxidative stress, accompanied by DNA damage and consecutive disorganized cell regeneration and apoptosis, can further exacerbate the disease and may lead to advanced fibrosis, cirrhosis and/or the development of hepatocellular carcinoma (HCC) in a minority of patients [3].

The underlying mechanisms leading to NAFLD development and progression are now considered to be complex and multifactorial [5]. Indeed, to describe the consequence of events driving the NAFLD pathophysiology, the “two hits hypothesis” was initially proposed [5]. According to that hypothesis, the rapidly growing consumption of high-fat diets in combination with the adoption of a sedentary lifestyle leads to obesity and insulin resistance, which culminate in hepatic accumulation of lipids, an event that acts as the first hit, further sensitizing the liver to other insults. These insults constitute the "second hit", which in turn activates inflammatory cascades and fibrogenesis [5]. However, this model became outdated since it was viewed as too simplistic to explain the underlying complexity of NAFLD pathogenesis, and, thus, it was replaced by the “multiple-hit hypothesis”, according to which the dietary habits and lifestyle, as well as environmental and genetic factors, can cause obesity, insulin resistance, ectopic adipose tissue accumulation and alterations in the intestinal microbiome, all of which are implicated in NAFLD pathogenesis and progression [5]. In this context, insulin resistance through the concomitant upregulated de novo lipogenesis in the liver and the reduced inhibition of lipolysis in the adipose tissue drives the upregulated transport of fatty acids to the liver that leads to steatosis/NASH, while fat accumulation further activates mechanisms related to mitochondrial dysfunction, oxidative stress, endoplasmic reticulum (ER) stress and production of reactive oxygen species (ROS), all of which leads to liver inflammation [5]. On the other hand, the genetic or epigenetic environment can further influence the fat content of liver cells, reinforcing the activation of the aforementioned mechanisms and can also affect several enzymatic procedures and the hepatic inflammatory status [5]. So, in conclusion, the mechanisms of insulin resistance, fat accumulation, mitochondrial dysfunction, oxidative stress, ER stress and ROS production, in association with genetic or epigenetic factors which can alter NAFLD predisposition, affect the fat content of hepatocytes, as well as hepatic pro-inflammatory pathways, culminating in chronic hepatic inflammation which can evolve into hepatocellular death, and activation of hepatic stellate cells that drive fibrogenesis [5].

Although significant efforts have been made over the last 40 years to elucidate the exact natural history and underlying biology of NAFLD, several challenges still exist [1]. The disease is under-recognized to a great extent by healthcare professionals as well as from society as a whole, while the lack of a trustworthy biomarker that would ideally diagnose NAFLD and its possible progression to NASH/HCC, renders invasive techniques, such as liver biopsy, indispensable for diagnosis, thereby inhibiting the early identification of persons in high risk [1]. Another challenge related to the elusive aspects of the underlying NAFLD biology is the significant heterogeneity of the disease and the currently restricted comprehension of its phenotypes [1]. Moreover, while several new drugs are being tested in clinical trials, so far, there are no approved therapies specifically for NAFLD/NASH, with most treatment guidelines focusing on lifestyle modifications with the adoption of a healthier lifestyle with weight loss and regular physical activity, and on the pharmacological treatment of comorbidities, such as anti-obesity and anti-diabetic drugs, and lipid-lowering drugs [1,3].

The need to fully decipher NAFLD pathogenesis and progression, as well as to conduct preclinical testing for potential therapeutic agents, has led to the need for reliable animal models of the disease that ideally display a liver phenotype similar to human NAFLD and can progress to inflammation, fibrosis (NASH), cirrhosis and HCC. Furthermore, these models should also display features of the metabolic syndrome, such as obesity and disturbances in lipid, glucose and insulin metabolism [6]. Towards this direction, several diet- or chemically-induced and genetic animal models have been introduced. Diet-induced models are mainly used in order to recapitulate a situation that mimics metabolic syndrome and to reproduce its main aspects, such as obesity and insulin resistance, that are crucially implicated in NAFLD development [7]. Due to that, the diet-induced models can better mimic the mechanisms, patterns and temporal sequence of events involved in human NAFLD pathogenesis. However, most of them differ from human disease in terms of clinical, morphologic and/or metabolic features [7]. Genetic models have been created to either mimic a human polymorphism implicated in NAFLD occurrence [such as the *patatin-like phospholipase domain-containing 3* (*PNPLA3*) polymorphic mice], to recapitulate characteristics of the human metabolic syndrome better than diet induction (such as leptin- or leptin receptor-deficient mice), to better depict or to more rapidly proceed to a particular stage of the NAFLD spectrum. These models can prove to be valuable for the investigation of specific molecular pathways, the mechanisms by which they can alter hepatic homeostasis contributing to NAFLD development and the consequences of their deregulation [8]. However, the genetic induction needed usually makes these mice different from humans who do not have these genes altered [8]. Moreover, many of the genetic murine models are monogenic, while NAFLD and its contributors, such as obesity, are multifactorial situations, and there is more than one route leading to its pathogenesis [9]. Chemically induced NAFLD mouse models are used to better study the progression from one disease stage to another. However, these interventions lead to artificial progression that does not recapitulate human etiology and pathology. Owing to the multifactorial nature of NAFLD, the combination of two or more inductions (diet, genetic or chemical) is a usual approach to better mimic human disease. The choice of the most appropriate model to be used depends on each particular researcher and study question. In this review, we present the key animal models that are currently utilized in each particular stage of NAFLD: (1) non-alcoholic fatty liver (simple steatosis-NAFLD), (2) NASH and (3) NASH-associated HCC.

## 2. NAFLD Mouse Models

### 2.1. Diet-Induced Models

#### 2.1.1. High-Fat Diet (HFD)

The increasing incidence of NAFLD in Western countries and its close association with obesity has led to the administration of a high-fat diet (HFD, developed to match Western diets) in order to lead to NAFLD development in animal models [10]. These HFDs range in fat content and source, with fat constituting 45% to 75% of the total calorie intake (kcal) and being derived from saturated fatty acids, polysaturated fatty acids and combinations of them [10,11,12]. These can lead to the development of metabolic syndrome, hepatic steatosis and NASH in experimental animals [13]. Samuels et al. were the first to implement such an HFD (fat constituted 80% of the total calories) in rats, which led to higher blood glucose levels observed at baseline and during exercise [14]. The most typical of these HFDs consists of 60% kcal as fat, 20% kcal as protein and 20% kcal as carbohydrates [6,15]. However, other such HFDs include a diet that contains 71% total calories (kcal) as fat, 11% as carbohydrate and 18% as protein (first HFD administered to rats inducing NASH) [16,17], or a diet consisting of 45% fat, 35% carbohydrates, and 20% protein [10]. Of note, HFDs fed to wild-type (WT) C57BL/6 mice can have varying effects depending on the time they are administered [6]. More specifically, when WT C57BL/6 mice are fed an HFD for 10–12 weeks, they develop steatosis, as shown by increased lipid accumulation, hyperlipidemia, hypercholesterolemia, hyperinsulinemia and glucose intolerance [10,15], whilst when the HFD is administered for 16 weeks then hepatocyte fat accumulation, ballooning, and Mallory-Denk bodies are observed, as well as decreased serum levels of the anti-inflammatory adipokine adiponectin and higher fasting serum glucose levels [6,17]. After feeding mice an HFD for 19 weeks, further hepatic triglyceride accumulation is observed, accompanied by inflammatory cell infiltration induction [15], while significant increases in circulating liver enzyme levels, i.e., alanine aminotransferase (ALT) and aspartate aminotransferase (AST), are observed after administration of the HFD for 34–36 weeks [6,15]. However, these mice show only minor signs of inflammation and fibrosis even after consuming an HFD for up to 50 weeks [6,15]. Interestingly, an HFD has also been administered as a chronic feeding scheme to C57BL/6 male mice for 80 weeks in order to mimic the lifetime consumption of a diet high in fat by humans [18]. This prolonged HFD feeding led to the development of obesity and insulin resistance, as well as to distinct histopathologic features of NAFLD, such as hepatic steatosis, cell injury, portal and lobular inflammation, fibrosis and hepatic ER stress [18]. Furthermore, this chronic HFD consumption caused changes in gut bacterial composition and led to gut-bacterial dysbiosis, a finding also observed in NAFLD patients [18]. HFD without other ingredients induced only mild steatohepatitis and minimal fibrosis, while the addition of lard enhanced steatosis and fibrosis [7]. In another study, C57BL/6 mice fed an HFD (58% of kcal as fat) supplemented with sucrose for 24–28 weeks displayed microvesicular steatosis and perisinusoidal focal hepatic fibrosis [19]. Overall, in HFD regimes, a higher percentage of cells enriched in fat is observed compared to other types of diets [10].

#### 2.1.2. High-Cholesterol Diet (HCD)

Dietary cholesterol has been recognized by several studies as a key contributor to the development of steatohepatitis in animal models and humans [10,20]. As such, a high-cholesterol diet (HCD) supplemented with 1% cholesterol has been fed to WT C57BL/6 mice [20]. Administration of this HCD leads to the development of simple hepatic steatosis with little inflammation and no fibrosis, as well as a marked increase in serum insulin levels [20]. However, liver weight, plasma triglyceride levels, free fatty acid levels, and serum ALT levels are similar or only slightly elevated compared to the control diet group [10,20]. The addition of cholesterol to this diet reduces bile acid and very low-density lipoprotein (VLDL) synthesis as well as the β-oxidation of fatty acids, mechanisms that could possibly contribute further to the hepatic lipid loading [20].

### 2.2. Genetic Models

#### 2.2.1. *db/db* Mice

Leptin is the prototype adipokine regulating—among other functions—the feeding behavior by reducing food intake through the promotion of satiety at the level of the hypothalamus [8,10]. *db/db* mice are homozygous for the autosomal recessive diabetic gene (*db*) [21], encoding a point mutation that leads to a lack of the long isoform of the leptin receptor (Ob-Rb), thus resulting in defective leptin signaling [8,10,22]. Consequently, these mice have normal or elevated levels of leptin, but are resistant to its effects [7,8], thus have persistent hyperphagia and are obese and diabetic, exhibiting severe hyperglycemia, hyperinsulinemia, insulin resistance, elevated serum leptin levels and development of macrovesicular hepatic steatosis [7,8,21]. Of note, *db/db* mice do not spontaneously develop inflammation when fed a normal control diet. However, calorie overconsumption for one month or longer could result in more aggravated hepatic inflammation [22]. In the absence of additional interventions, *db/db* mice are good models of NAFLD but not of NASH, since they rarely show NASH features when fed a normal control diet [10]. Despite that, when these mice are challenged with a “second hit” in terms of dietary interventions, they can develop NASH, as will be discussed in the next section [10,22].

#### 2.2.2. *ob/ob* Mice

The *ob/ob* mice were discovered by chance as a mutation arose in a colony at Jackson Laboratory in 1949 [23]. This mutation is recessive, causing sterility in the homozygotes, whilst these mice start becoming phenotypically distinguishable when they reach four to six weeks of age [23]. From that age on, they increase rapidly in weight until they reach a weight three to four times that of normal WT mice [23]. The *ob* gene was identified by positional cloning and was found to encode leptin [24,25]. By carrying this autosomal recessive mutation in the leptin gene, *ob/ob* mice have functional leptin receptors (in contrast to *db/db* mice), but they encode truncated and non-functional leptin [10]. Accordingly, *ob/ob* mice are obese, hyperphagic, inactive, hyperinsulinemic, hyperglycemic, hyperlipidemic, resistant to insulin, and develop intestinal bacterial overgrowth and spontaneous liver steatosis [7,10]. These mice develop fatty liver without fibrosis when fed a normal chow diet, and their resistance to hepatic fibrosis—unlike *db/db* mice—can be attributed to the requirement of leptin for fibrosis development [10,26]. Although *ob/ob* mice do not develop steatohepatitis, the latter can also be induced—like in *db/db* mice—with secondary hits/insults, such as diets other than chow feeding, exposure to low doses of lipopolysaccharide endotoxin [27], ethanol or hepatic ischemia/reperfusion challenge [10,28]

#### 2.2.3. *PNPLA3* Polymorphism

The PNPLA3 (patatin-like phospholipase domain-containing 3) protein is an enzyme with lipase activity towards triglycerides and retinyl esters and with acyltransferase activity on phospholipids [29]. The human *PNPLA3* gene encodes for a protein of 481 amino-acids, whilst the *PNPLA3* rs738409 variant is a cytosine to guanine substitution, which encodes for the isoleucine to methionine substitution at position 148 (I148M) of the protein [29]. The *PNPLA3* I148M variant in humans has been found to increase susceptibility to the whole spectrum of hepatic damage associated with NAFLD while it also promotes hepatic fat accumulation without directly affecting adiposity and insulin resistance to a significant extent [30]. Notably, it has also been associated with a higher risk of liver-related mortality in NAFLD patients and in the general population [30]. In order to mimic PNPLA3-induced fatty liver disease, transgenic mice in a C57BL/6 background were generated that express human PNPLA3 wild-type (*PNPLA3^WT^*) or the 148M variant (*PNPLA3^I148M^*) in the liver and adipose tissue [31], or the codon 148 of *Pnpla3* in the mouse genome was changed from ATT to ATG, substituting methionine for isoleucine (*PNPLA3^I148M^*) [32]. *PNPLA3^I148M^* mice with overexpression in the liver mimicked the fatty liver phenotype and other metabolic characteristics caused by this allele in humans and were presented with upregulated amounts of fatty acids and triacylglycerol [31]. Interestingly, when *PNPLA3^I148M^* mice were fed a high-sucrose diet (58% sucrose, 0% fat, or 74% sucrose) for 6 or 4 weeks, respectively, the hepatic triacylglycerol levels, as well as the levels of other lipids, increased up to 3-fold leading to steatosis development compared to *PNPLA3^WT^* transgenic mice or WT (non-transgenic) animals, while no such effect was observed under an HFD (45% kcal from fat) [31,32]. However, *PNPLA3^I148M^* mice display no changes in the expression of genes involved in triglyceride lipolysis, fatty acid oxidation, hepatic inflammation, or fibrosis after high-sucrose diet feeding [32].

#### 2.2.4. *Gankyrin* (*Gank*)-Deficient Mice

Gankyrin (Gank) is a subunit of the 26S proteasome and an oncogene expressed in several types of cancer. Moreover, it also drives liver proliferation and has recently been implicated in NAFLD development [19]. When *Gank* was selectively deleted from the livers of mice, *Gank* liver-specific knockout mice were generated and were administered an HFD (58% of kcal as fat) supplemented with sucrose for 24–28 weeks [19]. The livers of these mice displayed reduced proliferation and no fibrosis with strong macrovesicular steatosis, which was surprisingly associated with health improvement in the animals, while the livers of WT mice under the same diet developed fibrosis but with lower levels of steatosis [19].

## 3. NASH Mouse Models

### 3.1. Diet-Induced Models

#### 3.1.1. Methionine and Choline Deficient Diet (MCD)

The typical dietary NASH-inducing model is a diet consisting of high amounts of sucrose (40% of total calorie intake), moderately enriched in fat (10%), which also lacks the essential components of animal and human nutrition methionine and choline [10,11]. Methionine is an essential amino acid necessary for the synthesis of glutathione, an antioxidant protein, which also participates in the regulation of cell proliferation [33]. Choline is the precursor of phosphatidylcholine, which in turn is needed for VLDL secretion [34]. VLDL forms a lipoprotein complex that transports fatty acids from the liver to the adipose tissue [35]. Mice fed an MCD diet develop hepatic steatosis due to the high levels of fatty acid uptake and low levels of VLDL secretion, presenting hepatic inflammation after 2 weeks of feeding, as well as fibrosis after 6 weeks [10,11,36]. Plasma AST and ALT levels are increased, while lobular inflammation and ballooning degeneration of hepatocytes are also observed [10,37]. Furthermore, oxidative stress and an increase in pro-inflammatory cytokines/adipokines and pathways, such as nuclear factor-κB (NF-κB), occur in MCD-fed mice, which exacerbates hepatic damage [11,38,39].

#### 3.1.2. Atherogenic Diet (High Cholesterol and High Cholate)

An atherogenic diet typically contains high cholesterol and high cholate; cholesterol (1–1.25%) and cholate (0.5%) [7,40]. Increased plasma and liver lipid levels are observed in mice fed an atherogenic diet [40], which induces progressive steatosis, fibrosis, and inflammation, in a time-dependent manner ranging from 6 to 24 weeks. Furthermore, hepatocellular ballooning develops after 24 weeks on this diet [40].

#### 3.1.3. High-Fat (HF) Atherogenic Diet

The high cholesterol and high cholate atherogenic diet can be further supplemented with an additional high-fat component, such as 60% fat derived from cocoa butter [40]. Feeding mice this particular diet for 34 weeks leads to steatosis and NASH-like lesions, such as Mallory–Denk bodies and hepatocyte ballooning [7,41]. Furthermore, this diet worsens the histologic severity of NASH and induces insulin resistance and oxidative stress (downregulation of genes encoding antioxidant enzymes) whilst it further enhances the activation of hepatic stellate cells [7,40]. Interestingly, the HF atherogenic diet (60% fat, 20% carbohydrate, 20% protein, 1.25% cholesterol and 0.5% cholic acid) has been compared to the MCD diet (21% fat, 63% carbohydrate, 16% protein), and certain key differences between the two are noteworthy [6,42]. More specifically, while WT C57BL/6 mice fed both diets have high levels of hepatic free fatty acids, the MCD diet-fed mice mainly accumulate triglycerides in their livers, whereas the livers of HF-atherogenic diet-fed mice predominantly accumulate cholesterol [42]. Moreover, the administration of an HF-atherogenic diet does not cause a reduction in the weights of animal livers, while the MCD diet causes such a reduction [42]. Furthermore, compared to the HF-atherogenic diet, the MCD diet exacerbates the hepatic damage (e.g., as indicated by more increased circulating ALT and AST levels) and induces the formation of lipogranulomas (inflammatory nodules composed of macrophages surrounding a lipid droplet), and more pronounced fibrosis [42]. However, both the MCD and the HF-atherogenic diets are considered to induce the classic hepatic histopathological features of NASH, namely steatosis, hepatocyte ballooning and lobular inflammation to the same degree [42]. Additionally, transcriptional analysis has displayed that—although MDC and HF-atherogenic diets lead to the upregulation of hepatic stellate cell activation markers, pro-fibrogenic markers and extracellular matrix proteins—the expression of a subset of genes involved in extracellular matrix remodeling and production, as well as in hepatic stellate cell activation, is more dysregulated in MCD-fed mice than in atherogenic-fed mice [42]. Therefore, a more severe form of NASH is observed in MDC diet-fed mice, where extensive hepatic inflammation and fibrosis are observed quickly after the start of administration (2–10 weeks), probably due to the reduced VLDL synthesis and hepatic-β oxidation [6,42,43]. Notably, liver inflammation, steatohepatitis and elevated serum liver enzyme levels were normalized after switching the diet back to the control within 16 weeks, but fibrosis and CD68-positive macrophages remained present [44].

#### 3.1.4. High-Fat, High-Cholesterol (HFHC) Diets

NASH features are more striking when a high amount of cholesterol is combined with a high amount of fat or cholate in the diet [10]. The administration of a high-fat (33% kcal as fat) and high cholesterol (1% cholesterol) (HFHC) diet to WT C57BL/6 mice leads to greater weight gain, more pronounced hepatic lipid accumulation and steatosis, substantial inflammation, and perisinusoidal fibrosis (namely steatohepatitis) compared to the HCD [20]. This is also associated with adipose tissue inflammation [as indicated by high macrophage gene expression and increased gene expression of monocyte chemotactic factor genes and of the pro-inflammatory cytokine tumor necrosis factor α (TNF-α)], a reduction in circulating adiponectin levels, and a 10-fold elevation of serum ALT levels [20]. In addition, the HFHC diet also induces additional features of human NASH, including hypercholesterolemia and obesity [20].

In another regimen of high fat and cholesterol-enriched diet (containing 45% kcal fat and 0.12% cholesterol), WT C57BL/6 mice become obese and also develop hepatomegaly and hepatic steatosis, with varying degrees of liver fibrosis and steatohepatitis after consuming this diet for a long time (7 months) [45]. Furthermore, about 87% of the mice on this diet for 7 months have elevated plasma ALT [45].

In another study, the long-term consumption (for 20 weeks) of a high-caloric (43%) Western-type diet containing soybean oil with high n-6-polyunsaturated fatty acids (PUFA) (25 g/100 g) and 0.75% cholesterol induces steatosis, ballooning, inflammation and fibrosis, accompanied by hepatic lipid peroxidation, activation of Kupffer cells and oxidative stress in the liver of C57BL/6 mice [46]. Furthermore, the animals under this diet display increased weight gain and insulin resistance compared to those fed either a normal diet or a cholesterol-free HFD, thus showing a phenotype that reflects all clinical features of NASH in patients with metabolic syndrome [46].

In other cholesterol-enriched or choline-deficient high-fat diets, when C57BL/6 mice consumed a high-fat (23%), high-sucrose (424 g/kg) and high-cholesterol (1.9 g/kg) diet or a choline-deficient high-fat diet (CD-HFD) for three months developed marked steatohepatitis [10].

#### 3.1.5. High-Fat, High-Fructose (HFHF) Diets

Modern, western-style diets usually contain significant amounts of food rich in fructose, and this has been associated with a high incidence of obesity and NASH [47]. In order to mimic this pattern of nutrition, male WT C57BL/6 mice were fed a high-fat, high-fructose (HFHF) diet consisting of 58% kcal fat plus drinking water enriched with 42 g/L of carbohydrates at a ratio of 55% fructose and 45% sucrose (*w*/*v*) [47]. As a comparison, control animals were administered either a chow diet or an HFD containing 58% kcal fat to assess the effects of fructose consumption. Although weight gain, body fat, insulin resistance, hepatic steatosis, inflammation, and apoptosis were similar between the HFHF diet-fed and HFD-fed mice and increased compared to chow-fed ones, hepatic oxidative stress, pro-inflammatory monocyte populations (probably indicative of stellate cell activation), numbers of CD11b+F4/80+Gr1+ macrophages in the liver, transforming growth factor (TGF)-β1-driven fibrogenesis and collagen deposition were all elevated in the animals consuming the HFHF diet compared to those under the HFD [47]. These data suggest that fructose consumption is necessary for the progression of liver fat deposition to fibrogenesis [47]. In another study, C57BL/6 male mice were fed an HFHF diet (consisting of 60% of kcal as fat and 35% fructose) for 16 weeks, which, apart from the aforementioned increases in hepatic free fatty acid content, serum insulin levels, insulin resistance and oxidative stress, further led to hepatic iron overload, a finding often observed in humans with NAFLD [48].

#### 3.1.6. Amylin Liver NASH (AMLN) Model

Due to the variability in disease onset/progression and the lack of a preclinical mouse model that precisely reproduces the NAFLD-NASH phenotype, the Amylin Liver NASH (AMLN) model was developed [6,49]. The diet administered to these mice consisted of 40% fat (of which nearly 18% was trans-fat), 22% fructose, and 2% cholesterol, with the WT C57BL/6 mice that consumed this diet developing the three stages of NAFLD, namely steatosis, steatohepatitis with fibrosis, and cirrhosis, as assessed histologically and biochemically [49]. In these mice, the development of severe steatosis with necroinflammation and fibrosis, together with the appearance of periportal inflammation and hepatocellular ballooning, which are signs of progression to NASH, occurs after about 30 weeks on the AMLN diet [49]. However, the acceleration of this process and the development of more serious NASH is achieved by the combination of this dietary model with a genetic model. More specifically, when *ob/ob* mice were fed the AMLN diet for at least 8 weeks, key markers and histopathological features of NASH were induced [50,51]. This diet can act as a stimulus leading to NASH development in *ob/ob* mice since they cannot spontaneously proceed from NAFLD to NASH due to leptin absence, given that leptin is necessary for hepatic fibrosis development [26].

#### 3.1.7. Gubra Amylin NASH (GAN) Diet

Following the guidelines of the Food and Drug Administration (FDA) regarding the ban of trans-fats as food additives [52], there was a need for a non-trans-fat Western-type diet that would lead to NASH-like metabolic and hepatic histopathological changes comparable to those caused by the AMLN diet [53]. Thus, the Gubra amylin NASH (GAN) diet was developed with a composition and caloric density similar to the AMLN diet (40% kcal as high-fat, 22% fructose, 10% sucrose, 2% cholesterol) but with the substitution of trans-fat by saturated fat, i.e., palm oil (of the 40% high-fat, 0% was trans-fat and 46% was saturated fatty acids by weight) [53]. Notably, *ob/ob* mice fed the GAN diet for 16 weeks developed steatosis, fibrotic NASH (confirmed with biopsy), lobular inflammation, hepatocyte ballooning, fibrotic liver lesions and hepatic transcriptome changes, features similar to those induced in *ob/ob* mice on the AMLN diet and to those in patients with NASH [53]. The GAN diet induces fibrotic NASH in WT C57BL/6 mice as well, although, within a more prolonged time interval (28 weeks), similar to that needed by the AMLN diet to provoke the same NASH hallmarks [53]. However, the GAN-fed WT C57BL/6 mice displayed significantly higher body weight gain compared to those on the AMLN diet [53]. Moreover, C57BL/6 mice on the GAN diet for 38 weeks or more display morphological characteristics comparable to those in patients with NASH, with similar increases in markers of hepatic lipid accumulation, inflammation and collagen deposition as indicated by histomorphometric analysis. Furthermore, liver biopsies from GAN-fed WT mice and patients with NASH show comparable dynamics in several differentially expressed genes involved in key metabolic and histopathological features of NASH, such as nutrient metabolism, immune function and extracellular matrix organization [54]. Overall, the GAN diet is more obesogenic and impairs glucose tolerance compared to the AMLN diet [53].

#### 3.1.8. Fast-Food-like Diet (High-Fat/High-Fructose/High-Cholesterol Diet)

The administration of a different fast-food-like diet based on a different content of increased fat, fructose and cholesterol concentrations (41% fat, 30% fructose, 2% cholesterol) leads to NASH in different animal models, including the WT C57BL/6, *ob/ob* and KK-A^y^ mice; the latter being generated when the yellow obese gene (Ay) was transferred into KK-background mice, and exhibiting hyperphagia, obesity and hyperinsulinemia without fully progressing to NASH under normal chow diet [55,56]. Among these models, the *ob/ob* mice developed more marked NAFLD activity scores, fibrosis progression, obesity and hyperinsulinemia [55]. Interestingly, steatohepatitis- and fibrosis-related molecular pathways started displaying alterations after only two weeks of administering this fast-food-like diet in *ob/ob* mice [55]. In another study, C57BL/6 mice were fed an HFHF diet supplemented with high cholesterol as well (providing 40% of energy as fat, with 2% cholesterol and high-fructose corn syrup at 42 g/L final concentration administered in the drinking water of mice) which after six months led to the development of obesity, insulin resistance and steatohepatitis with pronounced ballooning and progressive fibrosis [57]. This was also accompanied by gene expression signature of increased fibrosis, inflammation, ER stress and lipoapoptosis, all key features of the metabolic syndrome and NASH with progressing fibrosis in humans [57].

### 3.2. Genetic Models

#### 3.2.1. *db/db* Mice with Iron Supplementation

When genetically obese *db/db* mice are fed a chow diet supplemented with high iron content (20,000 ppm vs. an iron content of 280 ppm in a normal chow diet), they develop a phenotype progressing from NAFLD to NASH [58]. Notably, this was characterized by hepatocellular ballooning, increased hepatic oxidative stress, inflammasome activation, impaired hepatic mitochondrial biogenesis and fatty acid β-oxidation, and activation of other pro-inflammatory/immune mediators [58].

#### 3.2.2. *foz/foz* Mice

Fat Aussie (*foz/foz*) mice bear a spontaneous deletion of 11bp in the *Alms1* gene, which is mutated in the Alström syndrome in humans, a disorder with distinct metabolic and endocrine features characterized by childhood-onset obesity, metabolic syndrome, and diabetes [59]. The ALMS1 protein localizes at the basal bodies of cilia, playing a role in intracellular trafficking, with the ALMS1-containing cilia in hypothalamic neurons being implicated in the control of satiety [59,60]. Accordingly, *foz/foz* mice fed a chow diet are hyperphagic, obese and glucose intolerant, with insulin resistance, decreased adiponectin levels, and increased total cholesterol, as well as increased hepatic weight, impaired hepatic function and steatosis [61]. Administration of an HFD (21–23% fat, energy content 19.4–20.0 megajoule/kg, 43% energy as fat) to *foz/foz* mice leads to severe NASH progression with significant upregulation of ALT levels, hepatocyte ballooning, inflammation, and fibrosis, as well as further decreased serum adiponectin levels, increased cholesterol levels, and higher hepatic triglyceride levels [59,62]. However, the characteristics and severity of diet-induced NASH are strain-dependent in *foz/foz* mice [63]. More specifically—while both *foz/foz* C57BL/6 and *foz/foz* BALB/c mice were similarly obese-hepatomegaly, hyperinsulinemia, hyperglycemia, and lower adiponectin levels occurred only in *foz/foz* C57BL/6 mice and not in *foz/foz* BALB/c mice after consuming either chow diet or HFD [63]. On the other hand, obese *foz/foz* BALB/c mice had more adipose tissue compared to *foz/foz* C57BL/6 [63]. HFD-fed *foz/foz* C57BL/6 mice present more severe NAFLD, as indicated by serum ALT levels, steatosis, hepatocellular ballooning, liver inflammation and NAFLD activity score, which were all higher compared to *foz/foz* BALB/c mice [63]. Of note, fibrosis after an HFD was severe in *foz/foz* C57BL/6 but absent in *foz/foz* BALB/c mice [63].

#### 3.2.3. MS-NASH Mice

The MS-NASH mouse (formerly known as FATZO/Pco) is a recombinant inbred cross of two strains prone to obesity when fed high-fat, high-calorie diets: C57BL/6J and AKR/J [64,65]. Crossing and selective inbreeding of these two strains lead to obesity, accompanied by significant insulin resistance, hyperlipidemia and hyperglycemia, as well as metabolic syndrome [66]. MS-NASH mice become obese and insulin resistant even when consuming a standard chow diet [64]. When these are fed a high-fat Western-type diet supplemented with 5% fructose in the drinking water for 20 weeks, they become heavier with higher body fat and hypercholesterolemia, with high AST, ALT, hepatic triglyceride levels, and liver over body weight [67]. Additionally, under the aforementioned Western-type, fructose-supplemented diet, markers of liver damage and evidence of NAFLD/NASH progression become more pronounced, particularly in male, and MS-NASH mice develop hepatic steatosis, lobular inflammation, hepatocellular ballooning and fibrosis [64,67].

#### 3.2.4. *Ldlr−/−* (Low-Density Lipoprotein Receptor Knockout)/*Ldlr−/−*.Leiden Mice

Deletion of the *low-density lipoprotein receptor* (*ldlr*) gene in mice has highlighted this gene as an important regulator of the transport of lipids and lipoproteins on macrophages and Kuppfer cells, opening a new field for NASH research [6,68]. Hyperlipidemic *ldlr−/−* mice are one of the two best characterized dyslipidemic models [69] and were first used as a model for atherosclerosis research, but when they were fed a high-fat diet with cholesterol (HFC) (21% milk butter, 0.2% cholesterol) for seven days the female mice developed steatosis with severe inflammation characterized by infiltration of macrophages and increased NF-κB signaling, while the male ones developed severe hepatic inflammation in the absence of steatosis [68]. Interestingly, a continuation of this diet for 3 months led to sustained hepatic inflammation in *ldlr−/−* mice, accompanied by increased apoptosis and hepatic fibrosis [70]. Furthermore, *ldlr−/−* mice under this HFC diet display high levels of low-density lipoprotein (LDL) and low levels of high-density lipoprotein (HDL), mimicking the corresponding human lipidemic profile [70]. Analogous to these findings, the administration of a “diabetogenic” high-fat diet supplemented with cholesterol (35.5% carbohydrate, 36.6% fat, 0.15% cholesterol) leads to obesity, insulin resistance, hyperinsulinemia, dyslipidemia, increased hepatic triglycerides and ALT, exacerbated hepatic macrophage infiltration, apoptosis, and oxidative stress, as well as to micro- and macrovesicular steatosis, inflammatory cell foci, and fibrosis in the livers of the *ldlr−/−* mice [71]. Another substrain of the *ldlr−/−* mice have been created with 94% C57BL/6 background and 6% 129S1/SvImJ background, namely the *Ldlr−/−.*Leiden mice (TNO, Metabolic Health Research, Leiden, The Netherlands) [72]. When *ldlr−/−.*Leiden mice are fed an HFD containing 45% kcal fat from lard and 35% kcal from carbohydrates (primarily sucrose) or a fast food diet (FFD) containing 41% kcal fat from milk fat and 44% kcal from carbohydrates (primarily fructose), they develop obesity, hyperlipidemia, hyperinsulinemia, increased ALT and AST levels, progressive macro- and microvesicular steatosis, hepatic inflammation, and fibrosis, along with atherosclerosis [72]. Interestingly, the HFD has been found to cause more severe hyperinsulinemia in these mice, while the FFD induces more severe hepatic inflammation with advanced bridging fibrosis, as well as more severe atherosclerosis [72].

#### 3.2.5. Prostaglandin E2-Deficient Mice

Although prostaglandins, and particularly prostaglandin E2 (PGE2), play a key role during pro-inflammatory processes [73], the role of PGE2 in liver inflammation is not fully elucidated [6,73]. To explore this, mice lacking microsomal PGE synthase 1 (mPGES-1), the key enzyme needed for the production of the majority of PGE2 during inflammation, were fed for 20 weeks a cholesterol-containing HFD with a high content of ω6-polyunsaturated fatty acids (PUFA) (255 g/kg fat, 0.75% cholesterol), which has previously been shown to induce NASH in mice [73]. Of note, these mice presented a strong infiltration of monocyte-derived macrophages and an increase in TNF-α expression in the liver, with the latter causing upregulated interleukin 1β (IL-1β) levels, primarily in hepatocytes and augmented hepatocyte apoptosis [73].

#### 3.2.6. *Apolipoprotein E2* Knock-In (APOE2ki) Mice

The *Apolipoprotein E2* (*APOE2*) knock-in (APOE2ki) mouse model was generated when the murine *Apoe* gene was replaced by a human *Apolipoprotein E2* (*APOE*2*) gene allele via targeted gene replacement in embryonic stem cells and was first used as a model for hyperlipoproteinemia and atherosclerosis [74]. When APOE2ki mice are fed a high-fat/high-cholesterol diet (42% energy as milk butter/fat, 0.2% cholesterol, 46% carbohydrates) from 7/10 days up to three months, they develop early (from the first seven days) increased inflammation indicative of NASH compared to C75BL/6 mice on the same diet which only develop simple steatosis and minor signs of an increase in the infiltrating inflammatory cells [70,75].

#### 3.2.7. *Apolipoprotein E*-deficient (*ApoE*−/−) Mice

Apolipoprotein E (ApoE) acts as a significant component of lipoprotein metabolism in humans and mice [76]. When it is absent, hypercholesterolemia, atherosclerosis and obesity have been reported to occur, thus *Apolipoprotein E*-deficient (*ApoE*−/−) mice display spontaneously increased inflammation and high levels of cholesterol compared to WT mice, and have traditionally served as a metabolic syndrome model used in cardiovascular research [76]. Indeed, *ApoE*−/− mice constitute the other most well-characterized dyslipidemic model [69]. However, when *ApoE*−/− mice consume a Western-type, cholesterol-enriched diet [42% energy as fat (with coconut oil), 1.25% cholesterol] for seven weeks, they show abnormal glucose tolerance, increased levels of fasting glucose, hepatomegaly, weight gain and develop the full spectrum of NASH spanning from hepatic steatosis and hepatocyte ballooning to inflammation and fibrosis [76]. Recently, *ApoE*−/− mice on HFD (with compositions ranging from 37–60% of kcal as fat, with a carbohydrate content of 38–44% and supplementation of 0.02–1.5% cholesterol and of 0.5% cholate, administered from seven to 24 weeks [77,78,79,80,81,82,83,84,85,86,87,88,89,90,91]) have been utilized as NAFLD/NASH models to investigate the pathogenesis and progression of this disease, as well as potential therapeutic treatments.

### 3.3. Chemically Induced Models

#### 3.3.1. Carbon Tetrachloride (CCl_4_) Administration

A way to explore the development of hepatic fibrosis is by inducing it with chemical agents, such as carbon tetrachloride (CCl_4_). When male BALB/c mice were given an intraperitoneal injection with CCl_4_ (0.4 mL/kg) twice a week for six weeks, their serum levels of aminotransferase and alkaline phosphatase were increased, whilst the anti-oxidative status of the liver was also disturbed [92]. The administration of CCl_4_ also caused extensive fibrosis [92]. Moreover, in a more recent model, male C57BL/6 mice were administered CCl_4_ and a Liver X receptor (LXR) agonist in combination with HFD feeding, which led to the development of insulin resistance and histopathological characteristics of NASH, such as macrovesicular hepatic steatosis, ballooning hepatocytes, Mallory-Denk bodies, lobular inflammation and fibrosis [93]. In this latter model, the TNF-α and interleukin-6 (IL-6) serum levels were significantly elevated, as well as the expression of mRNAs related to lipogenesis, oxidative stress, fibrosis and steatosis [93].

#### 3.3.2. Thioacetamide (TAA) Administration in Combination to Fast-Food Diet

Thioacetamide (TAA) serves as a hepato-toxin used to induce acute or chronic liver injury in mice and rats [94]. Combined administration of TAA (75 mg/kg, intraperitoneally, three times a week) with a fast food diet [12% saturated fatty acids (SFA) and 2% cholesterol with high fructose corn syrup (42 g/L final concentration)] to C57BL/6 mice for eight weeks leads to the appearance of key histological features of NASH, namely hepatic inflammation, hepatocellular ballooning, collagen deposition and bridging fibrosis [94].

### 3.4. Thermoneutral Housing

Apart from the genetic and dietary models trying to mimic the human NAFLD spectrum, a novel model focuses on the housing conditions of the animals and their effects on NAFLD development [95]. While mice are usually housed in a temperature range of 20–25 °C, this seems to be based on a human comfort zone and to stress the animals, while the thermoneutral zone or the temperature of metabolic homeostasis, for *Mus musculus* is 30–32 °C [95]. Wild-type C57BL/6 mice housed under thermoneutral conditions display exacerbated pro-inflammatory immune responses that were inhibited under standard housing conditions, and when they simultaneously receive an HFD (60% kcal from fat), they develop signs of deteriorating HFD-induced NASH, including increased steatosis, hepatocyte ballooning, exacerbated hepatic chemokine expression and macrophage infiltration of the liver, as well as more pronounced hepatic damage and elevated expression of genes associated with fibrosis, [95].

## 4. NASH-HCC Mouse Models

### 4.1. Diet-Induced Models

#### 4.1.1. Choline-Deficient L-Amino Acid-Defined (CDAA) Diet

The choline-deficient L-amino acid-defined (CDAA) diet is a modification of the MCD diet, in which proteins are substituted by an equimolar mixture of L-amino acids [7,11]. In WT C57BL/6 mice, this causes liver injury that mimics NASH features and can lead to the development of HCC [10]. More specifically, the CDAA diet reduces fatty acid oxidation in hepatocytes of mice, increases lipid deposition (especially after three months of treatment), induces oxidative stress and causes hepatic steatosis and inflammation (after six weeks) [96,97,98]. However, liver fibrosis develops gradually and takes quite a long time to occur: steatosis and lobular inflammation are induced after six weeks, slight features of fibrosis are displayed after 22 weeks, and finally, HCC is developed after about 36 weeks [96,97,98]. Although mice under the CDAA diet do not lose weight, they do not gain weight as well at least after 20 weeks on the CDAA diet compared to the control diet (choline-supplemented L-amino acid-defined diet), whilst their insulin sensitivity remains unchanged [99]. However, according to another study by Miura et al., insulin resistance sensitivity can be developed in CDAA-fed mice after 22 weeks, with obesity and increased plasma triglyceride and cholesterol levels, discrepancies probably attributed to the dietary compositions of the diet (e.g., percentage of fat) and the duration of the feeding [11,100].

#### 4.1.2. Choline-Deficient, Ethionine-Supplemented (CDE) Diet

Ethionine is a non-proteinogenic amino acid, the ethyl analog of methionine (an ethyl group instead of a methyl group) [11]. The choline-deficient, ethionine-supplemented (CDE) diet, which is derived from the MCD diet, induces steatohepatitis and inflammation shortly after the start of its administration, which is followed by fibrosis, cirrhosis and HCC in the long-term [11,101]. However, contrary to the NAFLD phenotype in humans, this diet leads to weight loss in CDE-fed animals, whilst it also causes high mortality [11,102,103]. To limit this mortality, some studies alternate the administration of CDE with a normal chow diet or combine a 100% CDE diet with a control diet to reduce the strength of the CDE diet to 75%, 70% or even 67% [101]. This approach minimizes morbidity and mortality while maintaining hepatic steatosis, inflammation and carcinogenesis [11,101].

#### 4.1.3. American Lifestyle-Induced Obesity Syndrome (ALIOS) Diet

In an attempt to recapitulate the western, modern lifestyle and its consequences on liver physiology, a frequently used diet is the American lifestyle-induced obesity syndrome (ALIOS) diet, which is based on the nutritional constitution of commonly consumed fast food, using WT mice kept in conditions that promote sedentary behavior [104]. Mice fed the ALIOS diet were consuming 45% calories in the chow from fat and 30% of the fat as trans-fat in the form of partially hydrogenated vegetable oil [28% saturated, 57% monounsaturated fatty acids (MUFA), 13% PUFA] and were also administered high-fructose corn syrup (HFCS) equivalent (55% fructose, 45% glucose by weight) in drinking water at a concentration of 42 g/L [104]. Furthermore, the cage racks were removed in order to prevent the physical activity of the ALIOS mice and mimic sedentary life [104]. After 16 weeks, the high trans-fat diet without the addition of fructose led to the development of a NASH-like phenotype with associated necroinflammatory changes, obesity, insulin resistance, and high plasma ALT levels accompanied by inflammatory and profibrogenic responses (as shown by the increased liver TNF-α and procollagen mRNA levels), as well as elevated levels of plasma insulin, resistin, and leptin [104,105]. The inclusion of fructose further increased food consumption, obesity, and impaired insulin sensitivity, while it did not alter the plasma ALT levels and the degree of steatosis [104]. The expression of some profibrogenic genes was upregulated, although no histologically detectable fibrosis was found in the mice, at least up to 16 weeks of ALIOS diet consumption [104]. A more prolonged administration of the ALIOS diet led Tetri et al. to the observations that mice display features of early NASH with mild or moderate steatosis at six months and characteristics of more advanced NASH at 12 months, including severe steatosis, liver inflammation, and bridging fibrosis, while Harris et al. reported that the characteristics of more advanced NASH are already present from 6.5 months [105,106]. Furthermore, increased expression of lipid metabolism genes (regulators of lipogenesis and β-oxidation) and insulin signaling genes was observed after six months of the ALIOS diet, which for some continued up to 12 months [106]. Notably, hepatocellular neoplasms were developed in 60% of the ALIOS-fed mice after 12 months, when there was also observed a marked expansion of murine hepatic stem cells, which were closely associated with the neoplastic foci [106]. The ALIOS diet has also been found to change the hepatic transcriptome of mice with the upregulation of genes associated with the reorganization of cellular structures, collagen binding, and inflammatory and immune responses, while genes associated with protein processing and metabolic procedures were downregulated [105].

#### 4.1.4. Diet-Induced Animal Model of Non-Alcoholic Fatty Liver Disease (DIAMOND)

The diet-induced animal model of non-alcoholic fatty liver disease (DIAMOND) was based on an isogenic strain that was derived from a cross between the common WT C57BL/6J strain and the 129S1/SvImJ strain, the latter being a commonly used model to create mice with targeted mutations [6,107]. Almost 60% of this strain’s genes come from the C57BL/6 background, while the isogenic nature of mice and the disease phenotype has been followed and confirmed over many generations [8,107]. These mice were fed a Western-type (high fat, high carbohydrate) diet consisting of 42% kcal from fat and 0.1% cholesterol, while they had ad libitum consumption of water with a high fructose and glucose content (23.1 g/L D-fructose plus 18.9 g/L D-glucose) [107]. These rapidly develop obesity, insulin resistance, hypertriglyceridemia, and increased levels of the total- and LDL-cholesterol, AST and ALT [107]. Following administration of this diet for longer periods, these mice gradually develop steatosis (4–8 weeks), steatohepatitis (16–24 weeks), progressive fibrosis (16 weeks onwards), with severe bridging fibrosis (by week 52) and spontaneous HCC (in 89% of mice between 32 and 52 weeks) [107]. Interestingly, DIAMOND mice display a pattern of pathway activation at the transcriptomic level similar to humans with NASH, with lipogenic, inflammatory and apoptotic signaling pathways being activated [107].

### 4.2. Genetic Models

#### 4.2.1. *Tm6sf2* Hepatic Knockdown or Knockout

Transmembrane 6 superfamily member 2 (TM6SF2) in humans regulates qualitative triglyceride enrichment in VLDL, lipid synthesis and the number of secreted lipoprotein particles [30]. The rs58542926 C > T polymorphism results in an A-to-G substitution in coding nucleotide 499 which replaces glutamate at residue 167 with lysine in the TM6SF2 protein (c.499A > G; p.Glu167Lys/E167K) [30,108]. The E167K variant of the *Tm6sf2* gene is strongly associated with hepatic triglyceride content and favors hepatic fat accumulation in intracellular lipid droplets since it decreases lipid secretion, leading to increased susceptibility to liver damage, including NASH and severe fibrosis, whilst it also predisposes humans to NAFLD with progression to HCC [30,108]. Furthermore, this variant has been associated with an increase in serum ALT, consistent with marked hepatic injury [108]. Selective knockdown of *Tm6sf2* levels in the livers of mice through the introduction of recombinant adeno-associated viral vectors expressing short hairpin RNAs (shRNAs) against *Tm6sf2* leads to 3-fold upregulation of hepatic triglyceride content, a significant decrease of plasma cholesterol levels and a tendency towards lower plasma triglyceride levels, findings consistent with a defect in VLDL secretion in mice consuming a chow diet *ad libitum* [108]. Feeding these mice with a high-sucrose fat-free diet (74% kcal from sucrose) for four weeks exacerbated the effects of *Tm6sf2* knockdown since levels of triglycerides and cholesterol esters were further upregulated in the liver and decreased in plasma [108]. Newberry et al. generated liver-specific *Tm6sf2* knockout mice, which developed spontaneous liver steatosis, with increased numbers of large lipid droplets, triglycerides and diacylglycerol species in the liver, as well as decreased hepatic VLDL triglyceride secretion under normal chow diet [109]. Moreover, *Tm6sf2* liver-specific knockout mice exhibited increased steatosis and fibrosis after the consumption of a high milk-fat diet (60.3% kcal from fat, primary milk fat) for three weeks, with those phenotypes being further exacerbated when mice were fed fibrogenic, high fat/fructose diets for 20 weeks [either a trans-fat, fructose supplemented diet with 45.3% kcal from fat, containing 22% hydrogenated vegetable oil and consuming sugar water containing 55% fructose/45% glucose (4.2g/L) or a palm oil diet (40% kcal fat) supplemented with 20% kcal fructose and 2% cholesterol] [109]. Finally, *Tm6sf2* liver-specific knockout mice are more susceptible to HCC since they display increased steatosis, greater tumor burden, and increased tumor area compared to non-transgenic floxed control mice when tumorigenesis is chemically/dietary induced [109]. Of note, a new rodent model was generated by Luo et al. by inactivating *Tm6sf2* in rats [110]. These *Tm6sf2*−/− rats display a 6-fold higher mean hepatic triglyceride content and lower plasma cholesterol levels than their WT littermates [110].

#### 4.2.2. Fatty Liver Shionogi (FLS) Mice

Inbreeding of dd Shionogi (DS) mice in 1984 resulted in some mice which developed spontaneous fatty liver without obesity, namely the Fatty liver Shionogi (FLS) mice [111]. These also display 5-fold higher hepatic triglyceride concentrations compared to DS mice and higher plasma AST and ALT levels, suggestive of hepatocellular lesions and inflammation [111]. After 2–4 months of age, mononuclear cell infiltration and clusters of foam cells appear in the fatty liver of FLS mice, accompanied by elevated serum ALT levels, suggesting the presence of NASH with inflammatory responses and liver injury [112]. Interestingly, FLS mice over 12 months may develop hepatocellular adenoma and/or HCC, with a frequency of 40% in males at 15–16 months and 9.5% in females at 20–24 months [112]. Interestingly, lipocalin-2 (LCN2), an adipokine, was overexpressed in the liver of FLS mice, particularly in hepatocytes localized around almost all inflammatory cell clusters, while the chemokine C-X-C motif ligand 1 (CXCL1), a pro-inflammatory chemokine exhibited hepatic overexpression localized in steatotic hepatocytes, and CXCL9 (another pro-inflammatory chemokine) was also overexpressed in hepatocytes and the sinusoidal endothelium localized in some areas of inflammatory cell infiltration [113]. When FLS mice were backcrossed to *ob/ob* mice, FLS-Lep*^ob^*/Lep*^ob^* or FLS-*ob/ob* mice were generated, which display remarkable hyperphagia, obesity, type 2 diabetes, severe hyperinsulinemia and hyperlipidemia, as well as histologically proven severe steatosis, inflammatory changes, increased oxidative stress with advanced fibrosis, and elevated levels of apoptosis later in life with spontaneous hepatic tumors [114,115].

#### 4.2.3. *Phosphatase and Tensin Homolog* (*PTEN*)-Deficient Mice

*PTEN* is a tumor suppressor gene encoding a multifunctional phosphatase, with both protein and lipid phosphatase activities, whose lipid phosphatase activity is associated with tumor suppression [116,117]. *PTEN* is mutated in multiple human cancers and is important for maintaining homeostasis and preventing oncogenesis in the liver, where it has been identified as a metabolic regulator of glycose metabolism and insulin signaling [117]. Hepatocyte-specific null mutation of *Pten* in mice has led to *PTEN*-deficient mice that display massive hepatomegaly and steatohepatitis with an accumulation of triglycerides, inflammatory cells and Mallory-Denk bodies, followed by liver fibrosis and HCC, closely mimicking the NASH phenotype/progression in humans [117,118]. These also develop insulin hypersensitivity [117]. Notably, liver tumors appear in 66% of the male and 30% of the female *PTEN*-deficient mice at 40–44 weeks of age and in 100% of PTEN-deficient mice at 74–78 weeks, with 66% of the tumors at 74-78 weeks being HCC (83% in male and 50% in female mice of that genotype) [117].

#### 4.2.4. *Augmenter of Liver Regeneration* (*Alr*) Hepatic Knockout (ALR-H-KO) Mice

Augmenter of liver regeneration (ALR) protein (encoded by the *Gfer* [*growth factor ERV1* homolog of *Saccharomyces cerevisiae*]) is pleiotropic and exhibits high expression in the liver (predominantly in hepatocytes), functioning as a hepatic growth factor which is critical for lipid homeostasis and mitochondrial function [119,120]. Global knockout of *Alr* in mice is embryonically lethal, so in order to investigate its role in the liver, mice with hepatocyte-specific deletion of *Alr* [*Alr* hepatic knockout (ALR-H-KO)] have been generated by Gandhi et al. [119]. These mice are normal at birth, but up to the second week of life, present low levels of ALR and ATP in their livers and have diminished mitochondrial respiratory function and increased oxidative stress, compared with livers from control mice, whilst also developing excessive steatosis and hepatocyte apoptosis [119]. Surprisingly, at weeks 2–4 after birth, hepatic lipid accumulation reverses, leading to a reduction in the levels of steatosis and apoptosis, while the numbers of cells that express ALR increase, along with ATP levels [119]. However, at 4–8 weeks after birth, and despite the reversion of steatosis, livers of ALR-H-KO mice develop hepatic inflammation, with hepatocellular necrosis, ductular proliferation, and fibrosis, whilst HCC appears in almost 60% of the mice by 12 months after birth [119]. When ALR-H-KO mice were administered a high-fat/high-carbohydrate (HF/HC) diet (60% kcal fat plus 2.3% fructose and 1.9% sucrose in drinking water) and were compared to WT and hepatocyte-specific ALR-heterozygous (ALR-H-HET) mice consuming the same diet, it was shown that the ALR-H-KO mice gained the least weight and had the least steatosis, whilst also had lower insulin resistance (all these HF/HC-fed mice developed insulin resistance) [120]. In addition, this HF/HC feeding led to severe fibrosis and/or cirrhosis in ALR-H-KO mice, contrary to the ALR-H-HET that only developed modest fibrosis and to WT that did not proceed to fibrosis and cirrhosis [120].

#### 4.2.5. *Melanocortin 4 Receptor* Knockout (*Mc4r*−/−) Mice

Melanocortin 4 receptor (MC4R) is a seven-transmembrane G protein–coupled receptor expressed in the hypothalamic nuclei, which regulates food intake and body weight [121]. In humans, *MC4R* mutations constitute the most common known monogenic cause of obesity [121]. Deletion of this receptor in mice results in the generation of *Melanocortin 4 receptor* knockout (*Mc4r*−/−) mice, which under a normal chow diet develop late-onset obesity, hyperphagia, hyperinsulinemia, and hyperglycemia [122]. *Mc4r*−/− mice consuming an HFD (60% energy as fat) for 20 weeks develop hepatic microvesicular and macrovesicular steatosis, ballooning degeneration, inflammation and pericellular fibrosis, a phenotype compatible with NASH, associated with obesity, insulin resistance, and dyslipidemia [121]. Overall, *Mc4r*−/− mice under HFD for 20 weeks develop both obesity and NASH with moderate fibrosis [123]. Interestingly, these mice when fed a western diet (41% kcal from fat, 43% kcal from carbohydrate, 17% kcal from protein) for 12 weeks, accompanied by intraperitoneal injection of lipopolysaccharide (LPS) twice a week for the last four weeks, develop NASH with rapid accumulation of fibrosis compared to HFD-fed *Mc4r*−/− mice [124]. Of note, *Mc4r*−/− mice further develop well-differentiated HCC after consuming HFD for 12 months [121].

#### 4.2.6. Tsumura-Suzuki Obese Diabetes (TSOD) Mice

Tsumura Suzuki obese diabetes (TSOD) mice spontaneously develop diabetes mellitus, obesity, glucosuria, hyperglycemia, and hyperinsulinemia without any special intervention/treatment, such as gene manipulation or an HFD [125]. The livers of TSOD mice display NASH-like features in the early stage at 3 months of age, which is exacerbated after 4 months of age, with microvesicular steatosis, hepatocellular ballooning, and Mallory-Denk bodies, with all these features worsening over time [125,126]. Interestingly, after 12 months of age, TSOD mice develop hepatic nodules with nuclear and structural atypia, characteristic of human HCC, with increased oxidative stress and elevation of glucose metabolites and L-arginine in the liver [125,127,128]. Although this model is not so common, it has been used over the last years in certain studies that investigated mechanisms implicated in NASH and hepatic carcinogenesis, as well as potential therapeutic agents against these conditions [129,130].

#### 4.2.7. *Keratin 18*-Deficient (*Krt18*−/−) Mice

Keratin 18 deficiency in mice (*Krt18*−/− mice) results in mild to moderate degree of steatosis after 17–20 months from birth, with lobular inflammation and mononuclear cell infiltration, an image that closely resembles NASH in humans [131]. These mice also develop liver tumors resembling human HCC with stemness features, frequently on the basis of chromosomal instability, at a frequency of ~80% and ~35% in male and female mice, respectively [131].

#### 4.2.8. *Methionine Adenosyltransferase 1A* (*Mat1a*) Knockout Mice

Methionine adenosyltransferases (MATs) are products of two genes, *Mat1a* and *Mat2a*, and catalyze the formation of S-adenosylmethionine (AdoMet), the principal biological methyl donor [132]. *Mat1a* is expressed in the liver, while *Mat2a* is expressed in extrahepatic tissues [132]. Mice with *Mat1a* gene deletion (*Mat1a* knockout mice) have increased hepatic weight compared to WT animals under a normal chow diet, while at eight months of age, they develop spontaneous macrovesicular steatosis and predominantly periportal mononuclear cell infiltration, signs of NASH [132]. Interestingly, these mice spontaneously develop HCC with increasing age (after 18 months of age) at a frequency of around 60% [133]. Notably, it has been now shown that the *Mat1a* gene deficiency impairs VLDL synthesis and dysregulates plasma lipid homeostasis, thereby contributing to the development of the NAFLD spectrum in these mice [134].

#### 4.2.9. Liver-Specific Deletion of *NF-κB Essential Modulator* (*Nemo*) in Mice (NEMO^LPC-KO^ Mice)

NEMO/IKKγ is a subunit of the IkB kinase (IKK), which is essential for the activation of the transcription factor NF-κB, thus regulating cellular responses to inflammation [135]. Deletion of NEMO in hepatic parenchymal cells leads to the development of steatosis and steatohepatitis characterized by immune cell infiltration of the liver parenchyma in 8-week-old NEMO^LPC-KO^ mice [135]. This is followed by inflammatory fibrosis and the appearance of dysplastic nodules at six months of age that culminate in the spontaneous development of HCC in 12-month-old mice [135].

#### 4.2.10. Circadian Rhythm Oscillations

Circadian rhythm has been found to exert multiple effects associated with NAFLD since it drives oscillations in mitochondria dynamics, oxidative stress, hepatic insulin resistance and triglyceride levels [136,137]. Liver-specific *Bmal1* knockout (LBmal1KO) mice accumulate oxidative damage and have been found to develop fatty liver and hepatic insulin resistance [136]. *Bmal1* knockout mice exhibit higher body fat than their WT littermates, impaired glucose metabolism, and high insulin sensitivity [138]. Notably, when these are fed an HFD (60% kcal fat) for 12 weeks, their insulin sensitivity is further increased, and they develop early obesity, to which they rapidly adapt [138]. In *Per1/2*−/− mice that lack a functional clock and consequently feeding rhythms, circadian oscillations in hepatic triglycerides levels persist, although with a completely different phase, pointing towards a role for additional mechanisms that control their circadian accumulation [137]. Interestingly, chronic disruption of the circadian rhythm has been found to disrupt the liver clock and induce NAFLD to NASH progression (high ALT and AST levels, hepatomegaly, chronic liver inflammation and fibrosis), which ultimately leads to HCC [139]. Furthermore, CLOCK mutant mice (*Clk*^Δ19/Δ19^) display steatosis and steatohepatitis with age when fed a chow diet compared to WT littermates (with differences being evident from 6 up to 12 months). Of note, their hepatic steatosis is accelerated when fed a cholate-containing high-fat diet (37% kcal from fat, 1.25% cholesterol and 0.5% cholate) and a Western diet (42% kcal from fat, 0.2% cholesterol), whilst they develop cirrhosis when challenged with LPS, which induces an inflammatory response, on top of a Western diet [140].

### 4.3. Chemically Induced Models

#### 4.3.1. Carbon Tetrachloride (CCl_4_) Administration in Combination with Western Diet

In a further attempt to accelerate the progression of extensive fibrosis and tumor development, a Western-type diet (high-fat, high-fructose and high-cholesterol, 42% kcal from fat, 41% sucrose and 1.25% cholesterol by weight) together with a high sugar solution (23.1 g/L D-fructose and 18.9 g/L D-glucose) was fed to WT C57BL/6 mice, while CCl_4_ at the dose of 0.2 μL (0.32 μg)/g of body weight was also injected intraperitoneally [141]. This approach resulted in the acceleration of the progression of the disease from simple NAFLD to NASH and tumor development, resulting in rapidly progressive steatosis, stage 3 fibrosis at 12 weeks, and HCC at 24 weeks [141].

#### 4.3.2. STAM Model

Streptozotocin (STZ) is an antibiotic that damages pancreatic islet β-cells and is widely used to generate a model of type 1 diabetes mellitus [142]. Indeed, STZ treatment causes insulin deficiency and hyperglycemia in mice and rats [142]. Accordingly, as an animal model to better understand the progression of NASH to HCC, the STAM mouse model was generated, in which neonatal WT C57BL/6 mice received a single low-dose (200 μg) STZ injection two days after birth followed by administration of an HFD (60% kcal from fat) after four weeks of age [143]. These mice display steatosis with first signs of lobular inflammatory accumulation at six weeks of age, marked inflammation with hepatocyte ballooning and progressive fibrosis at 8–12 weeks of age, and finally, HCC at 16–20 weeks of age [143].

## 5. Conclusions

The need for better animal models for the whole spectrum of NAFLD becomes even more relevant for translational research, as the NAFLD prevalence is increasing worldwide due to the increasing adoption of Western-type lifestyles and eating habits. Such models would advance our understanding of the molecular mechanisms that drive NAFLD pathogenesis, as well as facilitate the ongoing research for new NAFLD/NASH treatments. HFD feeding in mice is currently a widely used approach that, in comparison to other models, closely mimics the histopathology and pathogenesis of human NAFLD whilst it also presents important features that compose the broader clinical presentation of the disease. However, HFD alone can typically lead only to hepatic steatosis, and further combined interventions are needed to proceed to NASH and HCC in the utilized animal models. Moreover, HFD feeding is a very broad term that exhibits great variations regarding the utilized diet composition and duration of feeding, while it also seems to have different effects on mice of different genetic backgrounds. Similar issues, and especially the lack of a standardized diet regimen, are raised with other dietary interventions that lead to NASH, such as the high fat, high fructose (HFHF) diet or the fast-food-like diet (High-fat/high-fructose/high-cholesterol diet). Moreover, these diets and other NASH-inducing dietary models, such as the methionine and choline-deficient diet (MCD), either display discrepancies from the NASH in humans or require a long time to induce NASH features. Thus, most of the utilized NASH-inducing diets (e.g., the MCD, AMLN, GAN, and fast-food-like-diets) are usually administered to leptin-deficient (*ob/ob*) or leptin receptor-deficient (*db/db*) mice. Overall, the genetic alterations that have been introduced/utilized in mice models for NAFLD are rarely found in humans, or those that predispose to NAFLD in humans and are used to create mouse models (e.g., the *PNPLA3^I148M^* and *Tm6sf2* hepatic knockdown or knockout mice) do not seem to induce the same effects in the animal models. Therefore, so far, these models have certain limitations and fail to meet the features of an “ideal” NAFLD animal model. Furthermore, the genetic mouse models for NASH, such as the *Prostaglandin E2*-deficient mice, the APOE2ki and *ApoE*−/− mice, need a dietary stimulus as well to induce key NASH features. Finally, the administration of chemical agents, such as CCl_4_, TAA and STZ (in the STAM model), has the additional disadvantage of not reflecting the pathogenesis of NAFLD in humans.

Particularly, for research on NAFLD progression to HCC, the latter can be successfully caused by dietary modifications of the MCD diet (such as the CDAA or CDE diets), as well as by other diet regimes, but these models usually have discrepancies from what is noted in patients. Certain genetic models that have been created (*Krt18*−/−, *Mat1a* knockout and NEMO^LPC-KO^ mice) seem promising for the study of NASH and NASH-HCC since they display several key characteristics of the human disease. However, these have not been used in many studies yet, and thus remain to be further validated as appropriate models. TSOD mice seem another promising model that can develop all stages of NAFLD over time, with hepatic carcinogenesis being developed in a reasonable time range (after 12 months of age), whilst crucially these mice progress from NAFLD to NASH and HCC without the need for the addition of a genetic mutation/manipulation or a specific diet. Although not so common, this model has been used in some recent studies on the NAFLD spectrum [129,130], and its further use will reveal its exact potential as a more holistic NAFLD animal model.

On the other hand, there are also mouse models that can more properly replicate a specific stage of the NALFD but not its entire spectrum. For example, *ldlr*−/− mice exhibit an inflammatory response developing early within NASH. Thus, this model can serve to study the early onset of NASH [70]. Although not appropriate for mimicking the complete NAFLD human pathophysiology, these animal models provide the opportunity for research into a particular NAFLD stage and/or mechanism.

Given that the ideal animal model for NAFLD does not exist yet, the various animal models that are currently utilized for NAFLD research, with various dietary interventions, genetic manipulations, and administration of chemical substances and their combinations should be critically viewed within the context of their advantages and disadvantages for basic and translational research, as presented in Table 1. Accordingly, taking into consideration the specific research hypothesis/objectives of each study, there are models that can be successfully utilized in order to investigate the NAFLD-related pathophysiologic mechanisms and/or treatments.

## Author Contributions

Writing—original draft preparation, C.-M.F.; writing—review and editing, N.N.-A., I.K., B.M.L., M.L., A.C., G.K., E.K. and H.S.R.; supervision, E.K. and H.S.R. All authors have read and agreed to the published version of the manuscript.

## Figures and Tables

**Table 1 ijms-23-15791-t001:** Key advantages and disadvantages of animal models that are commonly utilized for NAFLD research, according to the stage of the disease they mimic.

	Advantages	Disadvantages
**NAFLD Mouse Models**		
*Diet-Induced Models*		
High-fat diet (HFD)	HFD-fed animals, in contrast to other NAFLD models, closely mimic both the histopathology and pathogenesis of human NAFLD [10]. Animals on a HFD display the most prominent features of NAFLD observed in humans as well, including obesity and insulin resistance [10].	HFD does not seem to be the best option to study NAFLD due to extensive variations related to dietary compositions (source, nature and composition of fatty acids in the diet), the age at which HFD feeding starts, duration of HFD feeding, gender and mouse strains (e.g., C57BL/6 mice seem to be susceptible to HFD effects, while BALB/c exhibit reduced hepatic lipid accumulation, maintain normal glucose tolerance and insulin action) [11,144,145]. Hepatic steatosis in HFD feeding varies in degree and seems to be dependent on various factors other than the diet, including the animal strain [10].
High-cholesterol diet (HCD)	HCD causes the development of simple hepatic steatosis and a striking increase in serum insulin levels [20].	HCD-fed mice display similar or only slightly increased liver weight, plasma triglyceride levels, free fatty acid levels, and serum ALT levels compared to the control diet fed animals [10,20].
	*Genetic Models*	
*db/db* and *ob/ob* mice	*db/db* and *ob/ob* mouse models develop characteristics and simulate many aspects of human metabolic syndrome, in contrast to various diet models [7,10]. Useful models of NAFLD, since they develop marked hepatic steatosis under the administration of a standard diet without additional hits [10]. They can develop steatohepatitis and be used to study NASH, as well, when a second hit is added, such as another type of diet [10].	Congenital leptin deficiency or resistance caused by leptin or leptin receptor mutations are extremely rare in NASH patients and leptin levels poorly correlate with the progression of simple steatosis to steatohepatitis, limiting the ability of *db/db* and *ob/ob* mice to reflect the etiology of human obesity, insulin resistance, and hepatic steatosis [7,10,146]. Leptin deficiency does not seem to play a role in NAFLD and NASH development in humans; on the contrary, serum leptin levels have been found elevated in NAFLD and NASH patients [147,148,149]. *db/db* and *ob/ob* mice consist monogenic animal models and the observations derived from them may not apply to humans in whom obesity is typically multifactorial [11].
*PNPLA3^I148M^* mice (PNPLA3 polymorphism)	*PNPLA3^I148M^* mice with *PNPLA3^I148M^* overexpression in the liver develop hepatic steatosis characterized by increased levels of triacylglycerol and other lipids [31]. *PNPLA3^I148M^* mice under high-sucrose diet present even higher triacylglycerol and fatty acid levels and more pronounced steatosis [31,32].	Hepatic fat levels remained unaltered and steatosis was not observed in *PNPLA3^I148M^* mice under HFD [31]. *PNPLA3^I148M^* mice develop diet-dependent hepatic steatosis, pointing towards diet as being the primary cause for *PNPLA3*-polymorphism-associated steatosis, thus rendering these mice a model that does not cover the full spectrum of NAFLD [6,32].
*Gankyrin* (*Gank*)-deficient mice	*Gank* liver-specific knockout mice with reduced liver proliferation develop pronounced macrovesicular hepatic steatosis, but no fibrosis after 6–7 months of HFD-feeding, in contrast to wild type littermates under the same diet that display lower steatosis and fibrosis [19]. This animal model led to the conclusion that liver proliferation drives fibrosis, while steatosis could probably play a protective role and thus potential NAFLD therapeutic approaches oriented towards inhibition of proliferation rather than inhibition of steatosis should be further investigated [19].	
**NASH Mouse Models**		
*Diet-Induced Models*		
Methionine and Choline Deficient diet (MCD)	MCD diet better emulates the pathological image and mechanisms implicated in the pathogenesis of NASH in humans than other dietary models [10]. Inflammation, fibrosis and apoptosis of hepatocytes developed more rapidly and severely than in mice fed an HFD or Western diets [10]. Oxidative stress, endoplasmic reticulum stress, and autophagocytic stress, three important cell stress-related mechanisms implicated in human NASH pathogenesis are significantly more active in MCD than in other dietary models [150]. Collectively, the MCD diet model is appropriate for studying histologically advanced NASH and the inflammation and fibrosis mechanisms in NASH [10].	Significant differences from the metabolic profile of human NASH exist in MCD diet: mice fed an MCD diet are not obese, but on the contrary show significant weight loss, cachexia, no insulin resistance, and low serum insulin, fasting glucose, leptin and triglyceride levels [10,151,152]. The main risk factors for the development of NAFLD and NASH in humans, namely increased weight and insulin resistance are lacking in this model [11]. These discrepancies limit the use of MCD diet models. Due to the aforementioned discrepancies, MCD diets are often fed to *db/db* or *ob/ob* mice to better mimic human NASH [10]. For example, *db/db* mice fed an MCD diet show marked hepatic inflammation and fibrosis [153]. Mallory-Denk bodies have not been observed in MCD diet-fed mice [7]. Different mouse strains display variable responsiveness to the MCD diet [10,11,154]. For example, C57BL/6N mice are more susceptible to NASH development when receiving the MCD diet in comparison with C3H/HeN mice [154].
Atherogenic diet	Atherogenic diet simulates some etiologic aspects of human NASH [7].	Weight loss, attenuation of insulin resistance and decreased serum triglyceride levels caused by atherogenic diet are not found in human NASH [7,40].
High-fat, high-cholesterol (HFHC) diets	Supplementation of dietary cholesterol and combination with high fat content triggers experimental hepatic inflammation and fibrosis and closely resembles clinical NASH features [20].	
High-fat, High-fructose (HFHF) diets	The obesity induced by HFHF diets appears to be a more robust model for human obesity in comparison with HFD [11]. HFHF diets appear to be a more robust model of NASH compared to HFD as well.	Lack of a standardized diet regime interferes with reproducibility [11].
Amylin Liver NASH (AMLN) model	The modifications made to the ALIOS model based on which the AMLN model was developed, namely the higher cholesterol content (2%) and the fact that fructose (22% by weight) was included in the food pellets rather than the drinking water, better mimicked the Western-type diet and the subsequent NASH occurrence [49].	NASH takes quite a long time to develop in wild type C57BL/6 mice after AMLN diet consumption [49], while acceleration of this process requires the administration of this diet to leptin-deficient *ob/ob* mice [50,51].
Gubra amylin NASH diet (GAN diet)	GAN diet is highly obesogenic and leads to the development of fibrotic NASH characteristics in *ob/ob* and wild type C57BL/6 mice, rendering it a suitable diet for preclinical therapeutic testing against NASH [53,54]. This diet induces a NASH image similar to that caused by AMLN diet, but with the substitution of trans-fats with saturated fats, following the latest FDA guidelines and by better replicating the Western type diet consumed by humans [53].	Fibrotic NASH takes longer to develop in wild type C57BL/6 mice after GAN diet consumption, compared to leptin-deficient *ob/ob* mice [53].
Fast-food-like diet (High-fat/high-fructose/high-cholesterol diet)	*ob/ob* mice consuming fast-food like diet develop metabolic, histological and transcriptomic characteristics similar to human NASH, and, thus, can serve as a preclinical model for testing drugs against NASH [55].	
	*Genetic Models*	
*db/db* mice with iron supplementation	Progression from NAFLD to NASH when genetically obese *db/db* mice are fed a chow diet with high iron content points towards a multifactorial role for iron overload in NASH pathogenesis in the setting of obesity and metabolic syndrome [58].	NASH development with a chow diet (even in the presence of excess iron) raises concerns about the translatability of this model [8].
*foz/foz mice*	*foz/foz* mice fed a chow diet are obese, hyperphagic, develop glucose intolerance, insulin resistance, decreased adiponectin levels, increased total cholesterol, and their livers have increased weight, impaired function and steatosis [61]. Administration of an HFD leads to severe NASH progression with major upregulation of ALT levels, hepatocyte ballooning, inflammation, and fibrosis, among other manifestations [59,62].	While both *foz/foz* C57BL/6 and *foz/foz* BALB/c mice become similarly obese after HFD feeding, *foz/foz* C57BL/6 develop more severe NASH, as indicated by higher NAFLD activity scores, fibrosis, hepatocyte ballooning and other necroinflammatory changes, implying that the severity of NASH in *foz/foz* mice is inconsistent and strain-dependent [63]. The exact role of *Alms1* gene is not fully understood and as a consequence the translational character of *foz/foz* model is rather limited [6].
MS-NASH mice	Unlike monogenic leptin deficient *ob/ob* or *db/db* mice, the MS-NASH mice are a polygenic model prone to obesity and type 2 diabetes when fed a standard chow diet, but have an intact leptin pathway, making this model more translatable to the human disease [67]. Given the metabolic status dysregulation when these mice are fed a Western-type, fructose-supplemented diet, this model can serve as a novel tool for studying NAFLD/NASH with high translational value [6].	Selective inbreeding can cause significant decrease in the genetic variability that may introduce a bias in preclinical drug testing [11].
*Ldlr−/−* (*low-density lipoprotein receptor* knockout)/ *Ldlr−/−*.Leiden mice	While most animal models for NASH imitate particularly the late stages of human disease, *ldlr−/−* mice focus on inflammatory response developing early within NASH, thus they can serve as a physiological model to study the early onset of NASH [70]. Although fibrosis is rather mild, the *ldlr−/−* mice develop more fibrosis compared to C57BL/6 mice consuming a similar high-fat diet supplemented with cholesterol [70]. *ldlr−/−*.Leiden mice can serve as a translational model of NASH with progressive liver fibrosis and simultaneous atherosclerosis development that can recapitulate human NASH, as indicated by hepatic transcriptome analysis [72]. Adaptation of the fat content of the HFD or fast food diet (FFD) administered to *ldlr−/−*.Leiden mice, can aggravate either the insulin resistance (HFD), or the hepatic inflammation and fibrosis (FFD) of the animals [72].	Sex differences are observed between *ldlr−/−*.Leiden mice when fed an HFD (45% kcal fat) for 18 weeks, with female mice being characterized by increased perigonadal fat mass, marked macrovesicular hepatic steatosis and liver inflammation and male mice displaying increased mesenteric fat mass, pronounced adipose tissue inflammation and microvesicular hepatic steatosis [155]. At young age, male *ldlr−/−*.Leiden mice are more susceptible to the detrimental effects of HFD than female ones [155].
Prostaglandin E2(PGE2)-deficient mice	Attenuation of PGE2 production by *mPGES-1* ablation in mice enhanced the inflammatory responses caused by TNF-α as well as hepatocyte apoptosis in diet-induced NASH [73].	The lack of genotype-specific differences in macrophage infiltration or fibrosis in PGE2 deficient mice under a cholesterol-enriched HFD denotes that duration of the feeding intervention (20 weeks) was probably not long enough to allow the development of more advanced stages of the disease [73].
APOE2ki mice	APOE2ki mice on an HFC (42% energy as milk butter/fat, 0.2% cholesterol, 46% carbohydrates) develop early-onset (from the first seven days) increased inflammation indicative of NASH [70,75].	The inflammatory response is not sustained in APOE2ki mice, as shown by the gene expression levels of inflammatory genes, and the numbers of infiltrating inflammatory cells that are reduced after three months of HFC diet [70]. Thus, probably the APOE2 isoform is not directly implicated in inflammation [70].
*Apolipoprotein E*-deficient (*ApoE*−/−) mice	*ApoE−/−* mice rapidly develop NASH within seven weeks of HFD usually—but not always—supplemented with cholesterol. In contrast, other dietary NASH models require a minimum of 15 weeks of diet or even longer induction time [76].	*ApoE−/−* mice spontaneously developed atherosclerotic plaques, while displaying some difference from humanized lipoprotein profiles, which suggests that this model is probably less suitable for human NAFLD research [6,76,156]. *ApoE*−/− mice show an obesity-resistant phenotype, resulting in remarkable insulin sensitivity, which is in contrast to what is noted for NAFLD in humans [157].
	*Chemically Induced Models*	
Carbon tetrachloride (CCl_4_) administration	CCl_4_ administration is an alternative way to study the development and progression of liver fibrosis [92]. Combined administration of HFD, CCl_4_ and a Liver X receptor (LXR) agonist led to the generation of an experimental model with histopathology and pathophysiology similar to human NASH [93].	The exact pathophysiological mechanism driving fibrinogenesis in the liver, and the role of hepatic stellate cells need further investigation [6].
Thioacetamide (TAA) administration in combination to fast-food diet	This model develops hepatic inflammation, fibrosis, and collagen deposition, hallmarks of human NASH and NASH-related liver injury [94]. The rapid development of these features in eight weeks makes this model promising for drug discovery investigation [94].	The mice of this model do not gain weight compared to control mice on chow diet and with no TAA injection, which is a significant discrepancy from human NAFLD/NASH [94].
	*Thermoneutral Housing*	
Thermoneutral housing	Thermoneutral housing combined with HFD leads to increased intestinal permeability and intestinal microbiome dysbiosis, features mimicking human NASH [95].	No signs of liver fibrosis were observed neither in male nor in female mice under HFD and thermoneutral housing, pointing towards the need for an additional dietary stimulus to induce liver fibrosis development [6,95].
**NASH-HCC Mouse Models**		
*Diet-Induced Models*		
Choline-deficient L-amino acid-defined (CDAA) diet	The combination of CDAA diet with HFD (CDAA + 60% kcal fat and 0.1% methionine by weight) can act as an improved CDAA model of rapidly progressive liver fibrosis, capable of reproducing the development of NASH, providing better understanding of human NASH and useful for the development of efficient therapies [6,98].	Humanized features of metabolic disturbances are missing in this model [6]. Due to low reproducibility among the different sources of CDAA diet, this model should not be used to study the metabolic profile of the disease [11].
Choline-deficient, ethionine-supplemented (CDE) diet	CDE diet induces steatohepatitis and inflammation in a relatively short period of time [11,101]. It recapitulates most of the stages of the spectrum of human NAFLD, progressing from liver damage to fibrosis, and finally HCC [102]. The fact that ethionine is also hepatocarcinogenic (produces hypomethylated DNA) makes this diet an interesting model for the study of steatohepatitis-related HCC [11,102,158].	CDE diet leads to weight loss in the animals to which it is administered and brings about high levels of mortality, reaching 60% after 4 months of feeding [11,102,103].
American lifestyle induced obesity syndrome (ALIOS) diet	The ALIOS diet when administered in mice mimics many of the clinical characteristics of NAFLD and, due to this reason, it serves as a robust and reproducible model for the investigation of NAFLD pathogenesis and progression [105].	The dietary composition of ALIOS diet differs from that normally consumed by humans, since the amount of trans fat per kilogram is greater than in commonly consumed fast food [6,104].
Diet-induced animal model of non-alcoholic fatty liver disease (DIAMOND)	DIAMOND mice when fed a Western-type diet (42% kcal fat, 0.1% cholesterol, 23.1 g/L D-fructose and 18.9 g/L D-glucose in drinking water) progressively mimic all key physiological, metabolic, histologic, transcriptomic and cell-signaling features of human NAFLD with progression to NASH and culminating in HCC after four months [107]. This model can serve as a preclinical model for the development of therapeutic targets of NASH [107].	A limitation of the DIAMOND model is that HCC develops in a high frequency, while in humans only a small proportion of NAFLD patients finally develops HCC [107]. DIAMOND mice do not develop neither significant atherosclerosis, nor fully established cirrhosis by week 52, features commonly seen in humans with obesity and NAFLD [107].
	*Genetic Models*	
*Tm6sf2* hepatic knockdown or knockout	*Tm6sf2* knockdown or knockout in mice livers upregulates hepatic triglyceride content and significantly decreases plasma cholesterol levels, leading to hepatic steatosis pointing towards a defect in VLDL secretion under normal chow diet, with these effects being exacerbated when those mice consume high-sucrose diets, leading to fibrosis development and even HCC after chemical/dietary tumorigenesis induction [108,109]. Therefore, knockdown of *Tm6sf2* selectively in mouse liver mimics the effects on hepatic triglyceride content and plasma lipids of the Glu167Lys TM6SF2 variant observed in humans [108].	*Tm6sf2−/−* mice replicate the human phenotype of the disease but are not very suitable for detailed mechanistic studies, a fact that led Luo et al. to the development of *Tm6sf2−/−* rats [110].
Fatty liver Shionogi (FLS) mice and FLS-*ob/ob* mice	FLS mice spontaneously develop hepatic steatosis, NASH after 2–4 months of age and liver tumors, including HCC after 12 months, although at a frequency less than 50% [111,112]. FLS-*ob/ob* mice are hyperphagic, obese with type 2 diabetes, hyperlipidemic and hyperinsulinemic, with severe steatosis in their livers, inflammatory changes, increased oxidative stress with advanced fibrosis, elevated levels of apoptosis later in life and spontaneous appearance of liver tumors [114,115]. Thus, it seems that these models can mimic the whole spectrum and progression of NAFLD.	The FLS mice (and not the FLS-*ob/ob*) display a very low rate of HCC incidence and heterogeneity of tumorigenesis; this model needs a long time to be established, and together with the fact that spontaneous HCC models are considered uncontrollable and unpredicted system, this model is rarely used [159].
*Phosphatase and tensin homolog* (*PTEN*)-deficient mice	*PTEN*-deficient mice develop hepatic steatosis, progressing to NASH and HCC development as they grow older up to 74–78 weeks of age, thus this model can prove useful for understanding NASH pathogenesis, progression from NASH to HCC and even for the development of potential NASH treatments [116,118].	These mice show insulin hypersensitivity and decreased body fat mass, characteristics that contrast with human NASH and limit the translational potential of the model [160].
*Augmenter of liver regeneration (Alr)* hepatic knockout (ALR-H-KO) mice	ALR-H-KO mice develop hepatic steatosis early in life, and although this is reversed between 2–4 weeks of age, they proceed to developing NASH, while HCC occurs at a frequency of about 60% after one year of life, thus they can provide a useful model for investigating the pathogenesis of steatohepatitis and its complications and can help in better understanding the progression from hepatic necrosis, inflammation and fibrosis to carcinogenesis [6,119].	
*Melanocortin 4 receptor* knockout (*Mc4r−/−*) mice	*Mc4r−/−* mice can serve as a NASH model used to investigate the sequence of events that lead to diet-induced hepatic steatosis, liver fibrosis, and HCC [121]. HFD-fed *Mc4r−/−* mice closely mimic the liver pathology and function of human NASH and could prove useful for the study of hepatic dysfunction during the fibrotic stage and generally advanced stages of NASH and of the potential effects of drugs on NASH development after HFD feeding [123].	
Tsumura-Suzuki Obese Diabetes (TSOD) mice	TSOD mice spontaneously develop NAFLD features, such as obesity, type 2 diabetes, hyperglycemia without any special treatment, diet or genetic manipulation, that quickly (after 3–4 months) proceed to NASH and lead to HCC after 12 months of age [125,126,127]. The NASH and HCC that the mice develop display many human-like characteristics, rendering this model a promising one for translational research of the NAFLD/NASH-HCC sequence and for the development of potential therapies [125,127].	
*Keratin 18*-deficient (*Krt18*−/−) mice	*Krt-18−/−* mice develop steatosis increasing with age, as well as progression to NASH and HCC that share several key characteristics with human disease, suggesting that this model can be used to dissect molecular pathways between NASH and HCC [131]. Interestingly, the male mice of this animal model display liver tumor formation at a higher frequency than female mice, as also observed in humans [131].	
*Methionine adenosyltransferase 1A* (*Mat1a*) knockout mice	*Methionine adenosyltransferase 1A* (*Mat1a*) gene deletion in mice impaired VLDL synthesis and plasma lipid homeostasis, thereby contributing to NAFLD development and spontaneous progression to NASH and HCC with increasing age [132,133,134].	
Liver-specific deletion of *NF-κB essential modulator* (*Nemo*) in mice (NEMO^LPC-KO^ mice)	Liver-specific deletion of *NF-κB essential modulator* (*Nemo*) leads to the occurrence of steatosis, NASH, inflammatory fibrosis and subsequently HCC with increasing age [135].	
Circadian clock oscillations	Chronic disruption of circadian rhythm can spontaneously lead to NAFLD progressing to NASH, fibrosis, and cirrhosis, similar to the human situation, pointing towards its translational value [6,139].	
	*Chemically Induced Models*	
Carbon tetrachloride (CCl_4_) administration in combination with Western diet	Combination of Western type diet, high-sugar drinking solution and CCl_4_ results in a mouse model with rapid progression of steatosis, advanced fibrosis and HCC, which also shares many histological, immunological and transcriptomic characteristics with human NASH, rendering it a possible useful experimental tool for preclinical drug testing [141].	The addition of CCl_4_ to Western diet results in blunting of weight gain and insulin resistance in those mice, typical features of NAFLD patients [141].
STAM model	Hepatic lipidemic profile of STAM mice shares many common characteristics with human NASH, despite the chemical intervention and the absence of obesity in this model [161]. NAFLD progression is an artificial process in this model and cannot accurately mimic the human disease pathology, limiting its preclinical potential [6].	This model represents NASH with diabetic background [143]. Weight gain and insulin levels of the STAM model mice are reduced compared to HFD-fed mice [161].

## Data Availability

Not applicable.

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
