# Peer review of "Genetic and Diet-Induced Animal Models for Non-Alcoholic Fatty Liver Disease (NAFLD) Research"

_ijms, 2022, doi:10.3390/ijms232415791_

Round 1
Reviewer 1 Report
The work is well organized and comprehensively described. The work is well organized and comprehensively described. The work is scientifically sound and not misleading. The references to related and previous work are appropriate.
Author Response
Response to Reviewer 1 Comments
Comments and Suggestions for Authors:
The work is well organized and comprehensively described. The work is well organized and comprehensively described. The work is scientifically sound and not misleading. The references to related and previous work are appropriate.
Response: We would like to thank the reviewer #1 for his/her encouraging comments and for his/her time for consideration of our manuscript.

Reviewer 2 Report
1. Can we revised the title to “Genetic and diet-induced animal models..)
2. Could you use non-breaking space between half of the word to incorporate it to the next line. For example, on line 81: Use Alt 8209 between guidelines and focusing to shift the word “focusing” to the next line. Please apply these to all such words throughout the review.
3. Could you include the type of animal model which demonstrates highest correlation between preclinical evaluations to clinical end points?
Author Response
Response to Reviewer 2 Comments
Comments and Suggestions for Authors
Point 1. Can we revised the title to “Genetic and diet-induced animal models..)
Response 1: We would like to thank reviewer #2 for his/her suggestion. The title is now changed to “Genetic and diet-induced animal models for non-alcoholic fatty liver disease (NAFLD) research”.
Point 2. Could you use non-breaking space between half of the word to incorporate it to the next line. For example, on line 81: Use Alt 8209 between guidelines and focusing to shift the word “focusing” to the next line. Please apply these to all such words throughout the review.
Response 2: We would like to thank reviewer #2 for his/her thoughtful comment. However, breaking of some words in two lines has been automatically created when we used the IJMS template. Our original manuscript contained no breaking words, however as we have to follow the International Journal of Molecular Sciences instructions and template, breaking of the words cannot be avoided.
Point 3. Could you include the type of animal model which demonstrates highest correlation between preclinical evaluations to clinical end points?
Response 3: We would like to thank reviewer #2 for his/her suggestion. However, it is not easy to be categorical and choose one model or one type of animal model that demonstrates highest correlation between preclinical evaluations and clinical end points. The reason for this is that each model or type of model has its own benefits and drawbacks and the preclinical evaluations and clinical end points are different in the various stages of the NAFLD spectrum (NAFLD, NASH or HCC). The choice of the most appropriate model to be used depends on each particular researcher and study question. The advantages and disadvantages of the types of animal models are now briefly discussed in the introduction (Page 3, lines 106-130):“Towards this direction, several diet- or chemically-induced and genetic animal models have been introduced. Diet-induced models are mainly used in order to recapitulate a situation that mimics metabolic syndrome and to reproduce its main aspects, such as obesity and insulin resistance, that are crucially implicated in NAFLD development [7]. Due to that, the diet-induced models can better mimic the mechanisms, patterns and temporal sequence of events involved in human NAFLD pathogenesis. However, most of them differ from human disease in terms of clinical, morphologic and/or metabolic features [7]. Genetic models have been created to either mimic a human polymorphism implicated in NAFLD occurrence [such as the patatin-like phospholipase domain-containing 3 (PNPLA3) polymorphic mice], to recapitulate characteristics of the human metabolic syndrome better than diet induction (such as leptin- or leptin receptor-deficient mice), to better depict or to more rapidly proceed to a particular stage of the NAFLD spectrum. These models can prove to be valuable for the investigation of specific molecular pathways, the mechanisms by which they can alter hepatic homeostasis contributing to NAFLD development and the consequences of their deregulation [8]. However, the genetic induction needed usually makes these mice different from humans who do not have these genes altered [8]. Moreover, many of the genetic murine models are monogenic, while NAFLD and its contributors, such as obesity, are multifactorial situations and there are more than one routes leading to its pathogenesis [9]. Chemically induced NAFLD mouse models are used to better study the progression from one disease stage to another, however these interventions lead to artificial progression that does not recapitulate human etiology and pathology. Owing to the multifactorial nature of NAFLD, the combination of two or more inductions (diet, genetic or chemical) is a usual approach to better mimic human disease. The choice of the most appropriate model to be used depends on each particular researcher and study question.” Furthermore, they have been also more extensively discussed in the conclusions section (Pages 18-19, lines 878-936).

Reviewer 3 Report
Manuscript review response: “Diet-induced and genetic animal models for non-alcoholic fatty liver disease (NAFLD) research”.
Manuscript ID: ijms-2040896
Comments
It is an interesting review that could provide relevant information about the best model to use to analyser to for non-alcoholic fatty liver disease in animal models, however, I have some questions about this manuscript
Line 73 can you mention how the mechanisms mitochondrial dysfunction, oxidative stress, endoplasmic reticulum (ER) stress and production of reactive oxygen species are associates with genetic or epigenetic factors?
Line 79 could you described What are the challenges that still exist on the underlying biology of NAFLD?
Line 93 it is necessary to write about the importance of genetic animal models and induced animal models because it is the central objective of the review in addition to the NAFLD, what advantages does one model offer over the other? and the importance of using one or the other? this depending on what you want to study.
Line 94 I suggest that you could write the 2 points only NAFLD and point 2.1 would be "induced model" because you could talk about mouses and rats or other animals and "High-fat diet (HFD)" it would you write as 2.1.1.
Line 145 you could show the point 2.3 as 2.2 “genetic model” and the point 2.4 as 2.3.1. you could repeat the same in the subsequent points.
Line 805 you could change animal models by animal induced models and animal genetic models.
Line 814 in this paragraph you could talk about the joint use of both diets, both induced or genetic or specify if you are talking about the induced or genetic model.
Line 818 you could write "animal model" instead of "mouse model".
it is necessary to homogenize the writing according to the title in talking about animal models instead of mouse models.
Line 855 It is necessary to separate the table in a section of induced models and a section genetic model and you could talk about
You could talk about the importance of using an induced model to that of a genetic model or vice versa.
Author Response
Response to Reviewer 3 Comments
Comments and Suggestions for Authors
Manuscript review response: “Diet-induced and genetic animal models for non-alcoholic fatty liver disease (NAFLD) research”.
Manuscript ID: ijms-2040896
Comments
It is an interesting review that could provide relevant information about the best model to use to analyser to for non-alcoholic fatty liver disease in animal models, however, I have some questions about this manuscript
Point 1: Line 73 can you mention how the mechanisms mitochondrial dysfunction, oxidative stress, endoplasmic reticulum (ER) stress and production of reactive oxygen species are associates with genetic or epigenetic factors?
Response 1: We would like to thank reviewer #3 for his/her comment. We now clarify in page 2, lines 72-86:“In this context, insulin resistance through the concomitant upregulated de novo lipogenesis in the liver, and the reduced inhibition of lipolysis in the adipose tissue, drives the up-regulated transport of fatty acids to the liver that leads to steatosis/NASH, while fat accumulation further activates mechanisms related to mitochondrial dysfunction, oxidative stress, endoplasmic reticulum (ER) stress and production of reactive oxygen species (ROS), all of which leads to liver inflammation [5]. On the other hand, the genetic or epigenetic environment can further influence the fat content of liver cells, reinforcing the activation of the aforementioned mechanisms and can also affect several enzymatic procedures and the hepatic inflammatory status [5]. So, in conclusion, the aforementioned mechanisms of insulin resistance, fat accumulation, mitochondrial dysfunction, oxidative stress, ER stress and ROS production, in association with genetic or epigenetic factors which can alter NAFLD predisposition, affect the fat content of hepatocytes, as well as hepatic pro-inflammatory pathways, culminating in chronic hepatic inflammation which can evolve into hepatocellular death, and activation of hepatic stellate cells that drive fibrogenesis [5].” how the mechanisms mitochondrial dysfunction, oxidative stress, endoplasmic reticulum (ER) stress and production of reactive oxygen species can act in combination with genetic or epigenetic factors to increase liver inflammation through their effect on the fat content of hepatocytes
Point 2: Line 79 could you described What are the challenges that still exist on the underlying biology of NAFLD?
Response 2: We would like to thank reviewer #3 for his/her comment. We now describe challenges that still exist on the underlying biology of NAFLD in page 2, lines 89-95:“The disease is under-recognized to a great extent from the healthcare professionals as well as from the society as a whole, while the lack of a trustworthy biomarker that would ideally diagnose NAFLD and its possible progression to NASH/HCC renders invasive techniques, such as liver biopsy, indispensable for diagnosis, thereby inhibiting the early identification of persons in high risk [1]. Another challenge related to the elusive aspects of the underlying NAFLD biology is the significant heterogeneity of the disease and the currently restricted comprehension of its phenotypes [1].”
Point 3: Line 93 it is necessary to write about the importance of genetic animal models and induced animal models because it is the central objective of the review in addition to the NAFLD, what advantages does one model offer over the other? and the importance of using one or the other? this depending on what you want to study.
Response 3: We would like to thank reviewer #3 for his/her valuable suggestion. We now discuss briefly the advantages and disadvantages of diet-induced, genetic or chemically induced animal models and the importance of using one type of model over the other, always depending on the specific study question of each researcher (Page 3, lines 106-130):“Towards this direction, several diet- or chemically-induced and genetic animal models have been introduced. Diet-induced models are mainly used in order to recapitulate a situation that mimics metabolic syndrome and to reproduce its main aspects, such as obesity and insulin resistance, that are crucially implicated in NAFLD development [7]. Due to that, the diet-induced models can better mimic the mechanisms, patterns and temporal sequence of events involved in human NAFLD pathogenesis. However, most of them differ from human disease in terms of clinical, morphologic and/or metabolic features [7]. Genetic models have been created to either mimic a human polymorphism implicated in NAFLD occurrence [such as the patatin-like phospholipase domain-containing 3 (PNPLA3) polymorphic mice], to recapitulate characteristics of the human metabolic syndrome better than diet induction (such as leptin- or leptin receptor-deficient mice), to better depict or to more rapidly proceed to a particular stage of the NAFLD spectrum. These models can prove to be valuable for the investigation of specific molecular pathways, the mechanisms by which they can alter hepatic homeostasis contributing to NAFLD development and the consequences of their deregulation [8]. However, the genetic induction needed usually makes these mice different from humans who do not have these genes altered [8]. Moreover, many of the genetic murine models are monogenic, while NAFLD and its contributors, such as obesity, are multifactorial situations and there are more than one routes leading to its pathogenesis [9]. Chemically induced NAFLD mouse models are used to better study the progression from one disease stage to another, however these interventions lead to artificial progression that does not recapitulate human etiology and pathology. Owing to the multifactorial nature of NAFLD, the combination of two or more inductions (diet, genetic or chemical) is a usual approach to better mimic human disease. The choice of the most appropriate model to be used depends on each particular researcher and study question.”
Point 4: Line 94 I suggest that you could write the 2 points only NAFLD and point 2.1 would be "induced model" because you could talk about mouses and rats or other animals and "High-fat diet (HFD)" it would you write as 2.1.1.
Response 4: We would like to thank reviewer #3 for his/her suggestion. We have now renumbered the sections, each section beginning with for example “2. NAFLD mouse models” or “3. NASH mouse models”, followed by “2.1” or “3.1” as diet-induced models and “2.2” or “3.2” as genetic models, and so on, and the specific diets or genetic interventions have been numbered “2.1.1”, “2.1.2” and so on. This has been applied to all subsequent points up to the “conclusions” section.
Point 5: Line 145 you could show the point 2.3 as 2.2 “genetic model” and the point 2.4 as 2.3.1. you could repeat the same in the subsequent points.
Response 5: We would like to thank reviewer #3 for his/her suggestion. We have now renumbered the sections, each section beginning with for example “2. NAFLD mouse models” or “3. NASH mouse models”, followed by “2.1” or “3.1” as diet-induced models and “2.2” or “3.2” as genetic models, and so on, and the specific diets or genetic interventions have been numbered “2.1.1”, “2.1.2” and so on. This has been applied to all subsequent points up to the “conclusions” section.
Point 6: Line 805 you could change animal models by animal induced models and animal genetic models.
Response 6: We would like to thank reviewer #3 for his/her comment. Since “Carbon tetrachloride (CCl4) administration in combination with Western diet” (line 805) and “STAM model” (line 814) constitute neither diet-induced nor genetic models, a new subsection “4.3. Chemically induced models” (Page 17, line 856) has been introduced and the aforementioned models now constitute the sub-subsections “4.3.1” (Page 17, line 857) and “4.3.2.” (Page 17, line 867) respectively.
Point 7: Line 814 in this paragraph you could talk about the joint use of both diets, both induced or genetic or specify if you are talking about the induced or genetic model.
Response 7: We would like to thank reviewer #3 for his/her comment. Since “Carbon tetrachloride (CCl4) administration in combination with Western diet” (line 805) and “STAM model” (line 814) constitute neither diet-induced nor genetic models, a new subsection “4.3. Chemically induced models” (Page 17, line 856) has been introduced and the aforementioned models now constitute the sub-subsections “4.3.1” (Page 17, line 857) and “4.3.2.” (Page 17, line 867) respectively.
Point 8: Line 818 you could write "animal model" instead of "mouse model".
Response 8: We would like to thank reviewer #3 for his/her comment. In line 818, only “model” has been used and not “mouse model”. However, the terms “animal model” and “mouse model” have been now added to make the sentence more clear to the readers.
Point 9: it is necessary to homogenize the writing according to the title in talking about animal models instead of mouse models.
Response 9: We would like to thank reviewer #3 for his /her suggestion. However, we cannot substitute the term “mouse model” with “animal model” throughout the whole manuscript, since we have to specify that this specific animal model (for example “STAM model”) is a mouse model and not a rat model. We have adopted the term “mouse model” where we thought that the term “animal model” could be misleading or confusing for the readers of our review.
Point 10: Line 855 It is necessary to separate the table in a section of induced models and a section genetic model and you could talk about
You could talk about the importance of using an induced model to that of a genetic model or vice versa.
Response 10: We would like to thank reviewer #3 for his/her suggestion. In accordance with the aforementioned renumbering and re-categorization of sections in the main text of the manuscript, each main section of the table, namely “NAFLD mouse models”, “NASH mouse models” and “NASH-HCC mouse models” has now been subdivided in sections of diet-induced, genetic and chemically induced models. Moreover, the corresponding diets, interventions and genetic models are included under each of the above sub-sections. The comparison between an induced and a genetic model is out of scope of this review since we have already discussed in several points of the main text the advantages and disadvantages of using a type of model over another. With this table, we aim to present the benefits and drawbacks of each one model and not of the diet-induced, genetic or chemically induced models collectively as categories.
Round 2
Reviewer 3 Report
I don´t have more questions